# VviPLATZ1 is a major factor that controls female flower morphology determination in grapevine

Pat Iocco-Corena[1], Jamila Chaïb[1,2], Laurent Torregrosa[3], Don Mackenzie[1], Mark R. Thomas[1] & Harley M. Smith [1✉]

Plant genetic sex determinants that mediate the transition to dioecy are predicted to be diverse, as this type of mating system independently evolved multiple times in angiosperms. Wild *Vitis* species are dioecious with individuals producing morphologically distinct female or male flowers; whereas, modern domesticated *Vitis vinifera* cultivars form hermaphrodite flowers capable of self-pollination. Here, we identify the *VviPLATZ1* transcription factor as a key candidate female flower morphology factor that localizes to the *Vitis SEX-DETERMINING REGION*. The expression pattern of this gene correlates with the formation reflex stamens, a prominent morphological phenotype of female flowers. After generating CRISPR/Cas9 gene-edited alleles in a hermaphrodite genotype, phenotype analysis shows that individual homozygous lines produce flowers with reflex stamens. Taken together, our results demonstrate that loss of *VviPLATZ1* function is a major factor that controls female flower morphology in *Vitis*.

[1] CSIRO Agriculture and Food, Locked Bag 2, Waite Campus, Glen Osmond, SA 5064, Australia. [2] Limagrain, HM.CLAUSE IBERICA S.A.U., La Mojonera Alméria, Spain. [3] AGAP & LEPSE, University of Montpellier, CIRAD, INRAe, Institut Agro, Montpellier, France. ✉email: harley.smith@csiro.au

Traits associated with flower development and fertilization are key for increasing yield in agriculture and horticulture crops[1]. Plants evolved multiple reproductive systems to promote outcrossing, including self-incompatibility, dioecy and dichogamy[2,3]. In a dioecy mating system, individual plants produce either male or female flowers that are morphologically distinct[4–6]. During plant speciation, the dioecy reproductive system independently evolved multiple times indicating that the underlying mechanisms that control female and male flower development are diverse. While approximately 5–6% angiosperms are dioecious, a number of economically important crops utilize this mating strategy[5–7].

In the past decade, one- and two- gene sex determination systems were characterized in a subset of dioecious plants[6]. A one-gene model of sex determination described in *Diospyros* and *Populus* is dependent upon the duplication of a single gene, which gives rise to small non-coding RNAs that alter flower sex identity[8–10]. *Actinidia* and *Asparagus* are two species that exhibit a two-gene model of sex determination involving DNA polymorphisms that results in: (1) ectopic expression of a male-promoting factor that suppresses pistil formation and (2) a recessive mutation in a single gene required for tapetum development and pollen fertility[11–14].

Grapevine cultivars derived from *Vitis vinifera* are a major perennial crop of economic significance throughout the world for wine, table and dried grape production[15]. Modern domesticated grapevines produce hermaphrodite flowers with functional stamens and pistils for self-pollination, which is a fundamental trait for productivity acquired during domestication[16]. Interestingly, the wild progenitor of *Vitis vinifera* grapevine cultivars, as well as other *Vitis* species, are dioecious[17]. In wild *Vitis* species, females initiate flowers that produce short reflex stamens that bend away from the receptive pistil, which likely reduces self-fertilization[18]. The reflex stamen phenotype is utilized by grapevine breeders and biologists to identify female vines for cultivar development. While pollen derived from female flowers is viable, fertility is absent due to an impairment in germination[19]. In males, flowers produce erect functional stamens, while pistil formation is aborted early in development, which prevents self-fertilization[19,20].

In *Vitis* species and hybrids, the *SEX-DETERMINING REGION* (*SDR*) is located on chromosome 2[21–25]. Depending on the *Vitis* species/hybrid accession and cultivar, the size of the locus varies from 111 kb to 837 kb; however, gene identity and content are conserved[26–29]. One hypothesis predicts that grape sex determination is mediated by the two-gene model[27,29,30]. According to this hypothesis, female flower formation is mediated by a recessive mutation that controls stamen development. Therefore, the genotype of female individuals is *f/f*. In contrast, male flower development is determined by a dominant male-promoting factor (*M*) that suppresses pistil development. Lastly, domesticated hermaphrodite grapevines with an *H/H* or *H/f* genotype produce perfect flowers that contain a modified and less dominant *M*-factor[27].

Through sequence analysis of *M*-, *H*- and *f*-haplotypes derived from domesticated and wild *Vitis* species, the structure of the *SDR* was determined and candidate sex-determining genes were identified[28,29,31,32]. Genetic association analysis of *M*- and *f*-specific DNA polymorphisms and phylogenetic studies of sex-linked genes indicates that male and female flower development is specified by genetic determinants located in the 5′ and 3′ region of the *SDR*, respectively[29,32]. The male sex-determining region (*MSDR*) consists of two genes, *Vitis vinifera YABBY3* (*VviYABBY3*) and *VviSKU5*, while 11 genes from *TREHALOSE-6-PHOSPHATE PHOSPHATASE* (*TPP*) to *WRKY21* reside in the female sex-determining region (*FSDR*)[32]. The *H*-haplotype is structurally similar to the *f*-haplotype from *VviYABBY3* to the aldolase gene but structurally related to the

*M*-haplotype from *TPP* to the *WRKY21* transcription factor[29,32]. Initial studies suggested that *VviAPT3* was the candidate *M*-factor that controls male flower formation through the inactivation of cytokinin[26,28,31,33], which is required for pistil development[34]. In support of these studies, transcript levels for *VviAPT3* are significantly increased in male flowers compared to hermaphrodite and female flowers[28,29,31] and the mRNA for this gene localizes to cells in the male flower meristem that gives rise to the pistil[33]. Consistent with the increased expression of *VviAPT3* in the male flower meristem, applications of cytokinin restore pistil development[35]. However, recent studies indicate that *VviAPT3* is not located in the *MSDR*[32]. Therefore, *VviYABBY3* emerged as the primary candidate *M*-factor[29,32] as this is one of two genes located in the *MSDR* and the expression of this gene associates with the formation of male flowers. Further, YABBY transcription factors are critical for the development of lateral organs, including carpels[36]. Genetic studies indicate that a candidate DNA polymorphism located in *Vitis INAPERTU-RATE POLLEN1* (*VviINP1*) is the underlying female sterility determinant, as this gene is predicted to play a role in pollen development[28,29,31]. While candidate flower sex genes have been identified from the above studies, functional analysis demonstrating a role for these genes in pistil and pollen development have yet to be determined.

Here, we show through genetic and molecular studies that the VviPLATZ1 (plant AT-rich sequence-and zinc-binding protein1) transcription factor is a candidate factor involved in female flower development. To this end, *VviPLATZ1* localizes to the *FSDR* and low expression of this gene correlates with the formation of reflex stamens. To validate the role of *VviPLATZ1* in reflex stamen development, the CRISPR/Cas9 system is used to generate multiple alleles of this gene lacking the conserved PLATZ domain in a rapid cycling hermaphrodite genotype. Phenotype evaluation shows that homozygous individuals harboring gene-edited alleles of *VviPLATZ1* produce flowers that initiate stamens with a reflex architecture. Taken together, results show that the formation of reflex stamens is mediated by the loss of *VviPLATZ1*, which is a factor involved in controlling female flower morphology in *Vitis*.

## Results

**Genetic and molecular analysis of *VviPLATZ1*.** As shown in Fig. 1a, association genetics indicates that the *FSDR* is comprised of 11 genes from *TPP* to *WRKY21*[32]. To further narrow down the number of genes that reside in the *FSDR*, 21 SNPs spanning the PN40024 *SDR* were identified (Supplementary Table 1). Three genetic mapping resources were genotyped with this *SDR* SNP set: (1) *Vitis vinifera* cultivars with known flower sex genotypes (*f/f*, *H/f* or *H/H*), (2) an $F_1$ 00C001V0008 (*f/f*) x Ugni Blanc (*H/f*) mapping population[37] and (3) self-fertilized hermaphrodite microvine (*H/f*) progenies (S1 to S5). However, as the PN40024 *SDR* contains a mixture of *H*- and *f*-sequences, this reference genome was not suitable for fine genetic mapping of the reflex stamen phenotype, as gene content between the *H*- and *f*-haplotypes differs in the *SDR*[27,31]. Genomic studies indicate that gene content and order in the *SDR* of *f*-haplotypes is conserved[28,29,32]. Therefore, we posited that the previously well characterized Cabernet Sauvignon *f*-haplotype sequence[29] was a suitable reference for fine genetic mapping of the *FSDR* by identifying the likely relative positions of the genic SNP markers developed in our study (Supplementary Table 1). By using this genetic mapping strategy, the genotype information derived from the *Vitis vinifera* cultivars indicated that the 5′ boundary of the *FSDR* was located in *TPP*, at the *VvMT_54* SNP (Fig. 1b; Supplementary Data 1). Genotype data from the $F_1$ 00C001V0008 (*f/f*) × Ugni Blanc (*H/f*) individuals further supported the location of the 5′ *FSDR* boundary in *TPP* (Supplementary Data 1). A single

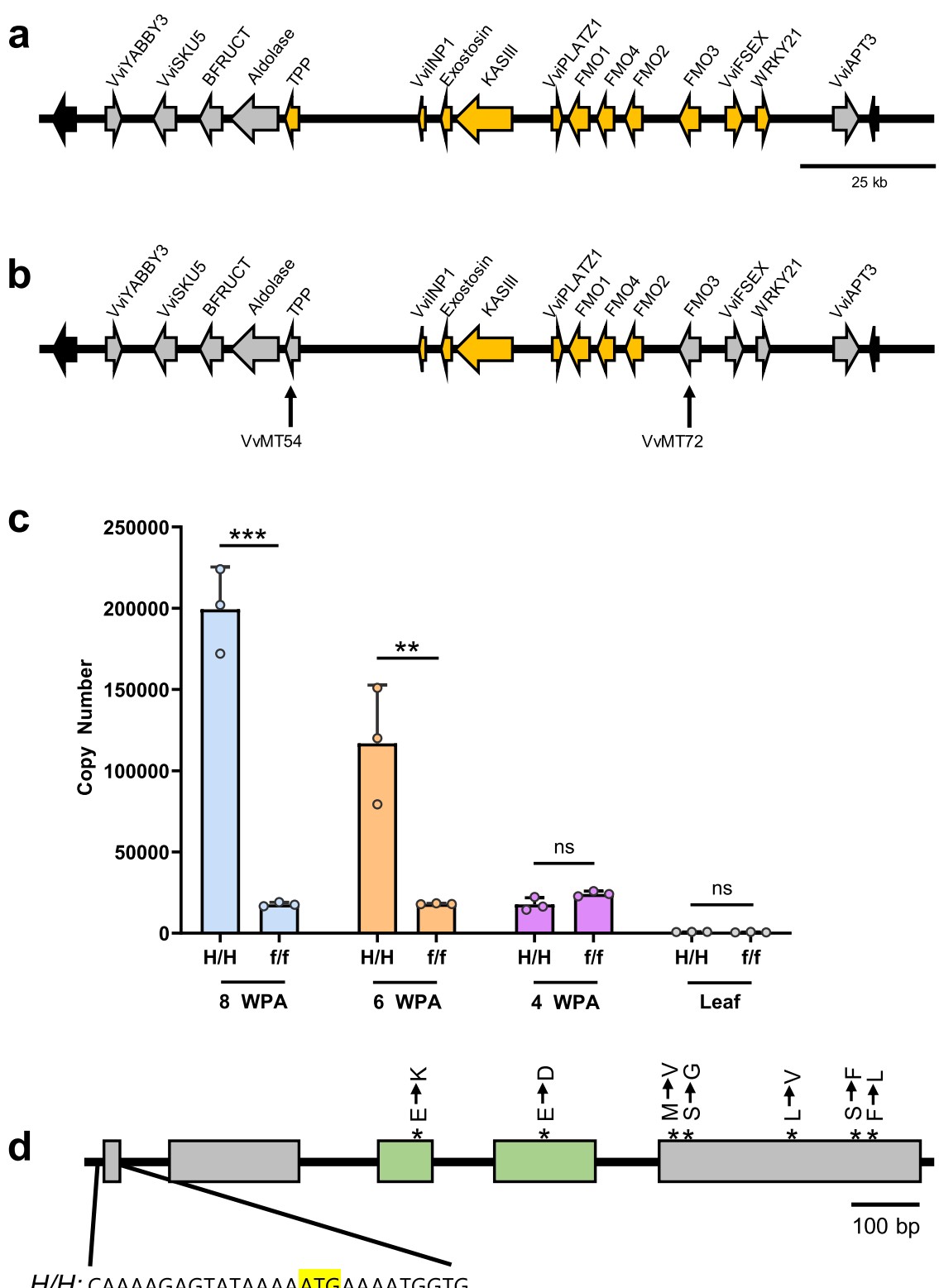

recombination event at *VvMT72* in the S5-microvine, 06C008V0018, defined the 3′ *FSDR* boundary located in *FLAVIN-CONTAINING MONOOXYGENASE 3 (FMO3*; Fig. 1b; Supplementary Table 1, Supplementary Data 1). Taken together, fine genetic mapping demonstrated that the *FSDR* contained seven genes: *VviINP1*, *Exostosin*, 3-ketoacyl-acyl carrier protein

synthase III (*KASIII*), *VviPLATZ1*, *FMO1*, *FMO4*, and *FMO2* (Fig. 1b).

The recessive mutation resulting in the formation of reflex stamens is predicted to reduce the expression and/or function of a candidate gene required for stamen development. Of the seven genes located in the *FSDR*, transcript abundance for *VviPLATZ1* appeared to correlate with the development of reflex

**Fig. 1 Structure of the *FSDR* and molecular analysis of *VviPLATZ1*. a** Diagram of the *SDR* structure derived from the Cabernet Sauvignon *f*-haplotype[29]. Genes localized within the *FSDR* are highlighted in yellow[32]. Gray arrows depict genes located outside of the *FSDR*. Black arrows were used to display genes flanking the *SDR*. **b** Fine genetic mapping narrowed the *FSDR* to seven genes, highlighted in yellow. The black vertical arrows indicate the location of the *VvMT54* and *VvMT72* SNPs in *TPP* and *FMO3*, respectively, which mark the boundaries of the *FSDR*. **c** Transcript levels determined for *VviPLATZ1* in 04C023V0006 (*H/H*) and 04C023V0003 (*f/f*) flowers at eight, six and four weeks prior to anthesis (WPA), as well as leaves (L). Note: stamen development and pistil initiation occurred at eight WPA[38]. Filament elongation and ovule development occurrs at six WPA. Stamen development is nearly complete by four WPA. Data are presented as mean values ± SD. Significant differences determined by two-tailed Student's *t* test indicated by asterisks **$p = 0.0089$, ***$p = 0.00027$ ($n = 3$); not significant, ns. **d** Diagram of *VviPLATZ1* genomic structure, which contains five exons (boxes). Exons 3 and 4 displayed in green encode the PLATZ domain. In the *f*-haplotype, DNA polymorphisms altered the nucleotides at −13 bp and −5 bp in the 5′ UTR, as well as the first codon (underlined nucleotides). The arrowhead and gray highlight illustrate the adenine insertion. The start codon for the hermaphrodite (*H/H*) and female (*f/f*) *VviPLATZ1* alleles is highlighted in yellow. Point mutations that alter amino acid content in exons 3, 4, and 5 are displayed by asterisk (*). The GenBank accession number for *VviPLATZ1* derived from 04C023V0006 is MW548436. For the *Vviplatz1* allele derived from 04C023V0003, the GenBank accession number is MW548435. Source data underlying Fig. 1c are provided as a Source Data file.

stamens[28,29,31]. Despite the results from these expression studies, a role for the loss of *VviPLATZ1* function in reflex stamen development was not recognized. Therefore, to support the hypothesis that loss of *VviPLATZ1* results in the formation of reflex stamens, transcript levels for this gene was examined in 04C023V0006 (*H/H*) and compared to 04C023V0003 (*f/f*) flowers at eight, six, and four week prior to anthesis (WPA), as previously defined[38]. Results showed that transcripts for *VviPLATZ1* were readily detectable in 04C023V0006 flowers at eight and six WPA (Fig. 1c), which correspond to a developmental time in which stamens differentiate and filaments elongate, respectively[38]. At four WPA, when flower development is nearly completed, the mRNA levels for this gene were significantly decreased. In addition, *VviPLATZ1* mRNA levels were readily detected in male flowers of 03C003V0060 (*M/f*) during stamen development at eight and six WPA (Supplementary Fig. 1). In contrast to 04C023V0006 and 03C003V0060, transcript levels for *Vvi-PLATZ1* were significantly reduced in 04C023V0003 during stamen development at eight and six WPA (Fig. 1c and Supplementary Fig. 1). Transcript levels for *VviINP1, Exostosin, KASIII, FMO1, FMO2,* and *FMO3*, which are located in the *FSDR*, did not correlate with the formation of reflex stamens[28,29,31].

DNA sequence comparisons between 04C023V0006 (*H/H*) and 04C023V0003 (*f/f*) identified female-specific DNA polymorphisms in *VviPLATZ1*. First, a thymine to adenine point mutation and adenine insertion were identified in the first codon, resulting in a frame-shift mutation that altered the position of the start codon (Fig. 1d). In addition, two SNPs were identified at −5 bp and −13 bp in the 5′ UTR. Through the alignment of hermaphrodite, male and female *VviPLATZ1* alleles from sequenced haplotypes of domesticated and wild *Vitis* species[29,31], results showed that female DNA polymorphisms at −13 bp, −5 bp, and +2 bp were conserved (Supplementary Fig. 2). However, the adenine insertion at +3 bp in the 04C023V0003 *VviPLATZ1* allele was found only in a subset of female-specific *VviPLATZ1* alleles. Taken together, the conserved female-specific DNA polymorphisms at −13 bp, −5 bp, and +2 bp may act to reduce transcription and/or alter mRNA decay of the female *VviPLATZ1* allele. In addition to these DNA polymorphisms, seven nonsynonymous SNPs were identified in the 04C023V0003 *VviPLATZ1* allele, which may reduce protein function, including a glutamic acid to lysine substitution in the PLATZ domain (Fig. 1d). To investigate whether these seven nonsynonymous SNPs were conserved, the 04C023V0003 VviPLATZ1 amino acid sequence was aligned with hermaphrodite, female and male VviPLATZ1 proteins derived from sequenced domesticated and wild *Vitis* species[29,31]. Results showed that all seven nonsynonymous SNPs were conserved in the female *VviPLATZ1* alleles (Supplementary Fig. 3). It should also be noted that the reduction in the number of transposable elements upstream of the female

*VviPLATZ1* allele compared to hermaphrodite and male alleles[28,31], as well as differences in RNA editing between male and female *VviPLATZ1* transcripts[39], may also be important factors attributed to low *VviPLATZ1* transcript accumulation in female flowers. As the reflex stamen phenotype is a recessive inherited trait, the female allele is referred to as *Vviplatz1*.

**Tissues-specific expression of *VviPLATZ1* during flower development**. To gain insight into the proposed function of *VviPLATZ1* in stamen development, the expression pattern of this gene was analyzed in 04C023V0006 (*H/H*) and 04C023V0003 (*f/f*) flowers during filament elongation at six WPA. At this time point, transcript abundance of *VviPLATZ1* was significantly higher in 04C023V0006 hermaphrodite flowers compared to 04C023V0003 female flowers (Fig. 1c). In hermaphrodite flowers, *VviPLATZ1* was primarily expressed in elongating filaments, as well as developing anthers and microspores (Fig. 2a–c). A similar pattern of expression was detected in flowers derived from the male microvine 03C003V0060 (*M/f*; Supplementary Fig. 4). In contrast, *VviPLATZ1* expression was relatively low in female filaments and anthers, as well as microspores (Fig. 2d–f). The expression of *VviPLATZ1* in ovules of hermaphrodite and female flowers (Fig. 2a, d, respectively), as well as male flowers (Supplementary Fig. 4) suggests that this gene may play a role in ovule development. Based on the molecular and gene expression studies described above, we propose that reduced *VviPLATZ1* function is a major factor that results in the development of reflex stamens.

**Gene-editing of *VviPLATZ1*.** Efficient validation of candidate genes implicated in flower development via reverse genetics is constrained by the perennial nature of grapevine, which displays a long juvenile phase and annual reproductive cycle[37]. The microvine model system has been utilized to overcome the genetic limitations of grapevine. This model system is highly amendable to rapid reverse genetics assessment, as the microvine can be stably transformed and displays a continuous flowering phenotype with a reduced generation cycle[37]. Therefore, the role of *VviPLATZ1* in stamen development was functionally validated in a homozygous hermaphrodite microvine genotype, 04C023V0006 (*H/H*), using the CRISPR/Cas9 system. Two guide RNAs designated FS1 and FS4 were developed to induce indels at the 3′ end of exon 2 upstream of the coding region for the PLATZ domain (Fig. 3a). Eight edited alleles were identified in $T_0$ plants, two of which contained a single bp insertion, while the remaining six alleles contained single or multiple bp deletions (Fig. 3b). All alleles altered the reading frame resulting in a truncated protein lacking the conserved PLATZ domain, which likely abolished VviPLATZ1 function (Fig. 3c). Interestingly, one of the $T_0$ plants

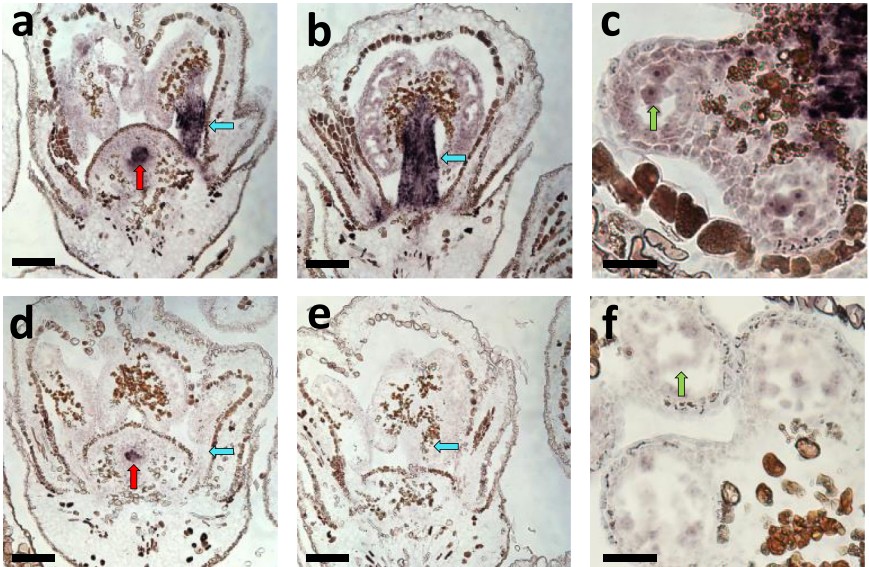

**Fig. 2 Expression pattern of *VviPLATZ1* in developing flowers at six WPA in hermaphrodite and female flowers. a–c** 04C023V0006 (*H/H*) flowers probed with *VviPLATZ1* RNA. *VviPLATZ1* expression was located in stamens (filaments and anthers; blue arrow), microspores (green arrow) and ovules (red arrow). **d–f** 04C023V0003 (*f/f*) flowers probed with *VviPLATZ1* RNA. Little or no *VviPLATZ1* RNA was detected in stamens (filaments and anthers; blue arrow) and microspores (green arrow). **a, d** *VviPLATZ1* was expressed in *H/H* and *f/f* ovules (red arrow). The representative images were derived from three biological replicas for 04C023V0006 and 04C023V0003. Scale bar is 100 µM for panels (**a**), (**b**), (**d**) and (**e**). For panels, (**c**) and (**f**), scale bar is 25 µM.

identified harbored both *Vviplatz1-1* and *Vviplatz1-2* alleles and was referred to as *Vviplatz1-1/2*. The other six $T_0$ individuals were heterozygous for only one of the remaining mutated alleles. All seven $T_0$ plants were self-pollinated and the flowers of homozygous $T_1$ *Vviplatz1-(1/2-8)* plants were analyzed and compared to hermaphrodite and female flowers.

**Loss of VviPLATZ1 function produced flowers with reflex stamens.** To determine if reduced *VviPLATZ1* function plays a role in female flower formation, stamen morphology was examined in hermaphrodite (*H/H*; *VviPLATZ1/VviPLATZ1*), female (*f/f*; *Vviplatz1/Vviplatz1*) and homozygous $T_1$ plants. Flowers produced in hermaphrodite individuals initiated upright stamens with an elongated phenotype (Fig. 4a, g). The average length of stamens in hermaphrodite individuals was 3.0 mm (Supplementary Fig. 5). In contrast, female flowers initiated reflexed stamens 98% of the time (Fig. 4b, g). The remaining 2% of female stamens displayed an upright growth pattern but were significantly shorter with an average length of 1.2 mm (Supplementary Fig. 5). Flowers produced in *Vviplatz1-3, -5, -7*, and *-8* displayed an overall morphology strikingly similar to female flowers (Fig. 4c–f, respectively). Flower morphology displayed in *Vviplatz1-1/2, -4*, and *-6* was similar to the structure of female flowers (Supplementary Fig. 6). In all seven *Vviplatz1-(1/2-8)* plants analyzed, 95 to 99% of stamens displayed a reflex architecture, while the remaining stamens were stunted with an average length of 1.4 mm (Supplementary Fig. 5). Taken together, loss of *VviPLATZ1* transforms hermaphrodites into females that initiate flowers with reflex stamens during flower development.

## Discussion

Dioecious mating systems evolved mechanisms to promote outcrossing through the arrested development of male and female reproductive organs and haploid sex cells in female and male individuals, respectively[6,7]. Floral morphology is a key factor that influences outcrossing, as exemplified by the spatial separation of anthers and stigmas in heterostyly flowers[40]. The formation of reflex stamens that spatially separates anthers from stigmas is a unique dioecious feature displayed by *Vitis* female flowers[18], which likely contributes to the outcrossing potential displayed in this dioecious mating system.

The formation of male and female flowers is influenced by hormones, which not only affects ovule and pollen development but also the morphology of floral organs, including petals, stamens, and pistils[41,42]. Experimental studies indicate that asymmetric distribution of auxin regulates organ bending during root and hypocotyl development[43]. In developing stamens, auxin biosynthesis, signaling and transport is critical for filament elongation[44,45]. Therefore, the symmetric distribution of auxin during filament elongation is predicted to cause organ bending for the reflex stamen phenotype[44,46]. It would be interesting to determine whether auxin is involved in the formation of reflex stamens and whether the distribution of this hormone is regulated by VviPLATZ1.

PLATZ genes encode plant-specific transcription factors that regulate a diverse array of reproductive processes including endosperm development, grain length and filling[47,48]. Experimental studies indicate that the *Zea mays floury3* (*fl3*) and *Oryza sativa GL6* regulate grain development through interaction with the RNA polymerase III (RNAPIII) complex, which transcribes transfer RNAs (tRNAs) and 5 s ribosomal RNA (5 s rRNA)[47–49]. Of the 17 PLATZ proteins identified in maize, 14 interact with a component of the RNAPIII complex indicating that PLATZ proteins regulate developmental processes via tRNA and 5 s rRNA biogenesis[49]. The role of VviPLATZ1 in stamen development further demonstrates that this class of plant-specific transcription factors regulates a wide range of reproductive processes. Further, it is tempting to speculate that loss of VviPLATZ1 function reduces the pool of tRNAs and 5sRNAs required for proper translation during stamen development. Lastly, phylogenetic analysis showed that VviPLATZ1 is a member of a clade that includes GL6, as well as uncharacterized PLATZ proteins from *Zea mays*, *Oryza sativa* and *Arabidopsis thaliana* (Supplementary Fig. 7).

While female flower morphology is determined by loss of *Vvi-PLATZ1* function, the *f*-haplotype-specific 8-bp deletion in the *Vitis*

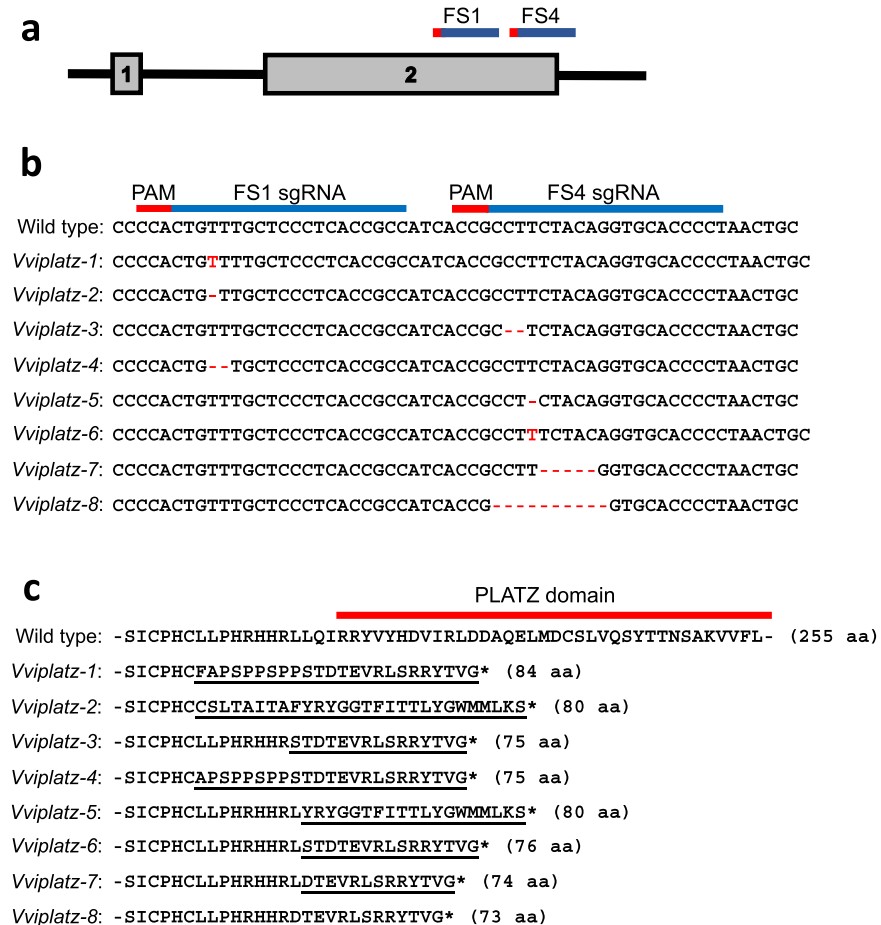

**Fig. 3 Molecular characterization of CRISPR/Cas9 gene-edited *VviPLATZ1* alleles. a** Schematic illustration of the *VviPLATZ1* genomic region targeted for CRISPR/Cas9 gene-editing. FS1 and FS4 guide RNAs were located at the 3′ end of exon two with FS4 traversing the exon two/intron two boundary. **b** Diagram of FS1 and FS4 guide RNA sequences aligned to *VviPLATZ1* and the resulting indels and base pair changes induced by CRISPR/Cas9 for each allele. FS1 and FS4 guide RNAs highlighted in blue, while the protospacer adjacent motif (PAM) is shown in red. **c** Diagram displaying the effect that each CRISPR/Cas9 induced mutation had on the protein sequence upstream of the PLATZ domain (red bar). Note: only the N-terminal region of the PLATZ domain is shown. Frameshift mutations alter protein sequence (underlined) resulting in truncated proteins smaller than the full length 255 amino acid (aa) protein.

*INAPERTURATE POLLEN1*-like gene (*VviINP1*) is a candidate mutation implicated in pollen fertility[28,29,31]. Indeed, inaperturate pollen is produced in female flowers of *Vitis riparia* and wild *Vits vinifera* individuals[19,50] While pollen fertility is not affected in Arabidopsis *inp1* plants[51,52], studies in maize indicate that *ZmINP1* is crucial for germination[53]. As a result, gene-editing of *VviINP1*, in grapevine, is necessary to validate a role for the 8-bp deletion in pollen fertility. However, VviPLATZ1 may also play a role in pollen function given its location within the *FSDR* (Fig. 1a, b), expression in microspores (Fig. 2c, f), and possible role in regulating the transcription of tRNAs and 5 s RNA, as protein synthesis appears to be a primary driver of pollen germination[54]. In support of this hypothesis, mutations in sex determination factors often display pleiotropic effects on other floral traits associated with unisexual flower development[55,56].

In conclusion, results from this manuscript show that VviPLATZ1 is a key regulator of female flower formation in grapevine. This is supported by the fact that *VviPLATZ1* is expressed in the filaments and anthers of hermaphrodite flowers prior to and at the time in which stamens elongate. Moreover, using the CRISPR/Cas9 gene-editing system in the rapid cycling hermaphrodite microvine, functional analysis demonstrated that loss of *VviPLATZ1* is a key factor that controls reflex stamen development during female flower formation.

## Methods

**Plant materials**. The Pinot Meunier L1 mutant, referred to as the microvine, contains a *GIBBERELLIC ACID INSENSITIVE1-like* (*VviGAI1*) gain of function allele. While *Vvigai1* vines display a dwarf phenotype[57], this mutation does not alter flower morphology[37]. The 04C023V0006 (*H/H*) and 04C023V0003 (*f/f*) microvines, which were derived from a Grenache (*H/f*) x L1 microvine cross (*H/f*), were used for molecular and gene expression analyses. Gene-editing was performed in 04C023V0006. Gene expression studies was also performed in 03C003V0060 (*M/f*), which was derived from 00C001V0008 (selfed L1 Pinot Meunier microvine) x Richter 110 (*V. berlandieri* cv. Boutin B x *V. rupestris* cv. du Lot). Plant material used for fine genetic mapping was derived from F1 00C001V0008 (*f/f*) x Ugni Blanc (*H/f*) individuals and successive self-pollinated microvines (*H/f*) from first (S1) to fifth generation (S5)[37]. These mapping lines were maintained in glasshouses and growth rooms under long-days (16 h light/8 h dark) with 27 °C and 22 °C day and night temperatures, respectively. *Vitis vinifera* cultivars used in this study were derived from the cultivar collection at the CSIRO vineyard in Merbein, VIC, Australia.

**DNA marker development and fine genetic mapping of the *FSDR***. Sequences for the SSR markers, UDV027, VVIB23, VMC3B10 and VMC6F1, flanking the previously identified *SDR*[22–24,58], were used to identify the genomic region of this locus in the 8X PN40024[59] and Pinot Noir[60] genomes. The genomic sequences of annotated genes regularly spaced between UDV027 and VMC6F1 in PN40024 and Pinot Noir were scanned for candidate SNPs using the basic local alignment search tool (https://blast.ncbi.nlm.nih.gov). Thirty-three candidate SNPs were identified in exons and/or introns of selected genes in the region from VIT_202s0025g04920 to VIT_202s0154g00230 (Supplementary Table 1). The Agena Bioscience MassARRAY platform (Agena Bioscience, San Diego, California, USA) was used to validate the 33 SNPs by genotyping *Vitis vinifera* cultivars (*n* = 33) and self-pollinated

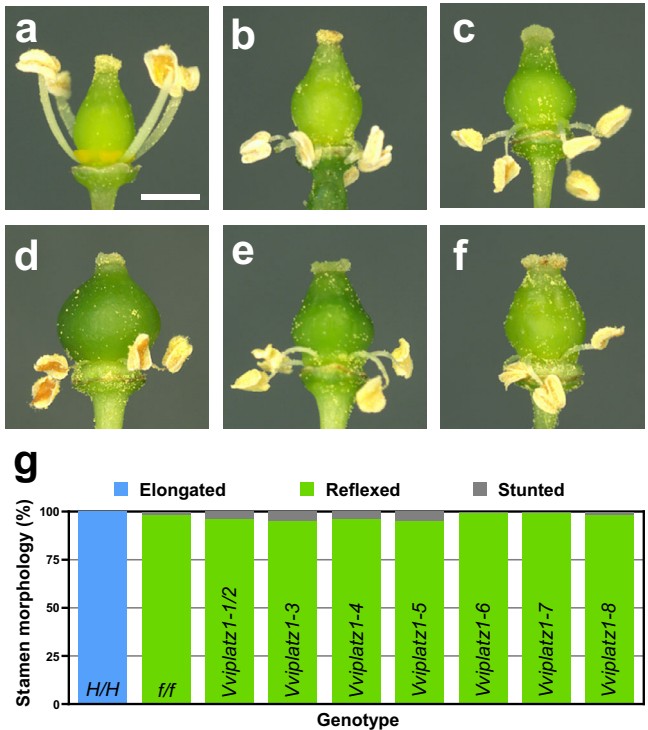

**g**

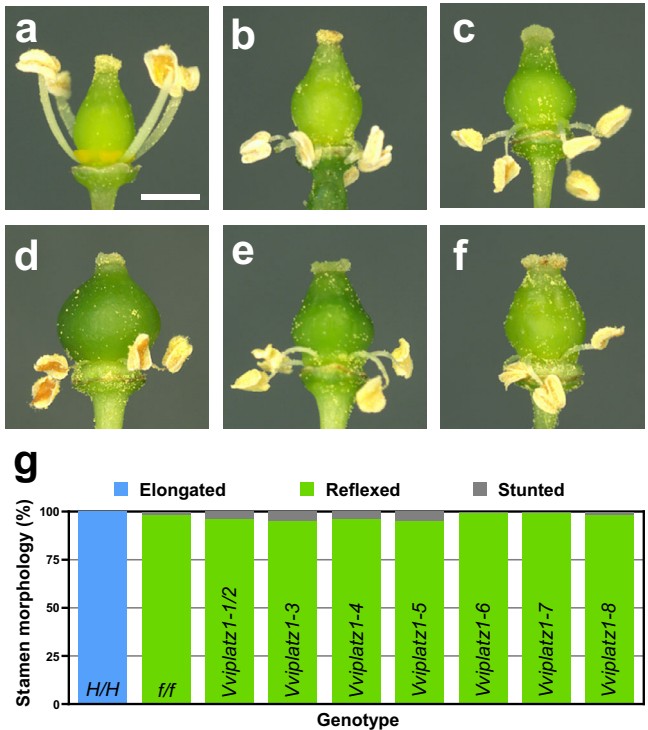

**Fig. 4 Reflex stamen development is controlled by loss of *VviPLATZ1* function.** Representative images of flowers derived from (**a**), 04C023V0006 (*H/H; VviPLATZ1/VviPLATZ1*); (**b**) 04C023V0003 (*f/f; Vviplatz/Vviplatz*); (**c**) homozygous *Vviplatz1-3*; (**d**) *Vviplatz1-5*; (**e**) *Vviplatz1-7*; (**f**) *Vviplatz1-8* plants. **g** Quantification of stamen architecture in 04C023V0006, 04C023V0003, and homozygous mutated plants. Elongated stamens (blue) with an upright growth pattern and an average length equal to 3.0 mm was initiated in *H/H* flowers. Reflex stamens that bend away from the pistil were produced in *f/f* and homozygous mutated flowers (green). In addition, 1–5% of the stamens initiated in *f/f* and homozygous mutated flowers were stunted and grew to an average length of 1.2 and 1.4 mm, respectively. The scale bar is 1.0 mm and is representative for all flower images (**a**–**f**). Source data underlying Fig. 4g are provided as a Source Data file.

microvines (*n* = 37) outlined in Supplementary Data 1. In addition, the F₁ 00C001V0008 x Ugni Blanc mapping population (*n* = 101) was also genotyped with this SDR SNP set. The MassARRAY genotyping was performed at the Australian Genome Research Facility (AGRF; https://www.agrf.org.au/genotyping). In this procedure, DNA was isolated from immature leaves using the NucleoSpin® 96 Plant II DNA extraction kit (Macherey-Nagel, Düren, Germany, #740663.4). PCR was used to amplify the 33 SNP-targeted regions in each of the DNA samples. After dephosphorylating excess nucleotides in the PCR reactions, a single base extension reaction was performed using an extension primer for each of the 33 SNPs. The extension reaction utilized a single mass-modified dideoxynucleotide complementary to each of the SNP alleles. Each SNP allele was identified by the distinct mass of the extension primer using MALDI-TOF mass spectroscopy analysis[61]. Allele calling was performed using the Agena Bioscience MassARRAY software, version 4.1.0.83. After MassARRAY genotyping, 21 SNPs were validated for fine genetic mapping of the *FSDR* (Supplementary Table 1). The PCR primers used to amplify the 21 SNP-targeted regions, as well as the corresponding extension primers are shown in Supplementary Table 2.

To perform the fine genetic mapping, the positions of the 21 validated SNPs were identified in the *SDR* sequences of the Cabernet Sauvignon *H*- and *f*-haplotypes. This was achieved by aligning a 50 bp PN40024 genomic sequence containing each SNP to the Cabernet Sauvignon *H*- and *f*-haplotypes using the basic local alignment search tool (Supplementary Table 1). Sequence and gene organization of the Cabernet Sauvignon *f*-haplotype together with the genotype information shown in Supplementary Data 1 was used to delineate the boundaries of the *FSDR*.

**RNA extraction and RT-qPCR.** Total RNA was extracted from 100 mg of flowers derived from at least three inflorescences from a single 04C023V0006 (*H/H*),

04C023V0003 (*f/f*) and 03C003V0060 (*M/f*) plant at eight, six and four WPA (Supplementary Fig. 8)[38], using the Spectrum Plant Total RNA Kit (Sigma-Aldrich Pty. Ltd., Sydney, NSW, Australia, #STRN250). After RNA extraction, samples were purified using the RNA Clean & Concentrator™ (Zymo Research, Irvine, California, USA, #R1013). Total RNA was also extracted and purified from 100 mg of leaf tissue derived from 4 immature leaves, approximately 25 cm² in size, from 04C023V0006, 04C023V0003, and 03C003V0060. Three biological replicates were used for quantifying *VviPLATZ1* transcript abundance in developing flowers at eight, six and four WPA, as well as leaves, for 04C023V0006 and 04C023V0003. For 03C003V0060, only two biological replicates were used for determining *VviPLATZ1* expression in developing flowers at eight, six, and four WPA, as well as leaves (Note: each biological replicate was derived from a single plant). First strand cDNA was synthesized from 1 µg of total RNA using the SuperScript™ IV Reverse Transcriptase (Thermo Fisher Scientific, Waltham, Massachusetts, USA, #18090010) at 53 °C for 1 hr. After denaturation at 80 °C for 10 min, the cDNA samples were diluted 1:10 in H₂O. RT-qPCR was performed with the LightCycler® 480 SYBR Green I Master mix (Roche Diagnostics, Mannheim, Germany, #04887352001) using 2.5 µl of diluted cDNA sample with gene-specific primers for *VviPLATZ1* (VvPla_RT_F: CCCCTGTTTCTCTCCGAACT) and VvPla_RT_R: GCGCTTTTTCTTCACGAACT), *VviUbi*[62] and *VviActin2*[63] with a final concentration of 0.75 µM. Amplification was performed using the Roche LightCycler 480 system with the following parameters: 5 min at 95 °C, 20 s at 59 °C and 20 s 72 °C (45 cycles), 5 min at 72 °C, followed by a melt cycle of 15 s at 95 °C, 45 s at 50 °C, continuous heating to 95 °C at 11 °C s-1[64]. To calculate DNA copy number, the amplified PCR product for *VviPLATZ1*, *VviUbi* and *VviActin2* was cloned into Qiagen pDRIVE vector using the Qiagen PCR Cloning Kit (Qiagen, Chadstone, Victoria, Australia, #231222). After sequencing the clones, the DNA concentration (ng/µl) for each construct was determined using the Quanti-iT™ PicoGreen® dsDNA Quantification Kit (Promega, Madison, Wisconsin, USA, #E2670) and the GloMax® Discover Microplate Reader (Promega Madison, Wisconsin, USA, #GM3000). DNA copy number was calculated in each standard dilution[65]. After amplification, a standard curve derived from the absolute DNA copy number values from the dilution series for each gene was used to calculate transcript copy number for *VviPLATZ1*, *VviUbi* and *VviActin2* in each of the samples using the Roche LightCycler 480 software. Transcript copy number for *VviPLATZ1* was normalized against reference genes, *VviUbi* and *VviActin2* for each sample.

**Cloning *VviPLATZ1* alleles.** To clone *VviPLATZ1*, the B26 primer (GACTC-GAGTCGACATCGATTTTTTTTTTTTTTTTTT) was used to prime cDNA synthesis in the 04C023V0006 (*H/H*), 04C023V0003 (*f/f*) and 03C003V0060 (*M/f*) using cDNA samples derived from flowers at six WPA. Subsequently, *VviPLATZ1* mRNA was amplified using primers CSPla1_CDS_F1 (CAGTGCCAGTTTTG-CAGGC) and B25 (GACTCGAGTCGACATCGA). After subcloning the PCR products into the pDRIVE vector using the Qiagen PCR Cloning Kit (Qiagen, Chadstone, Victoria, Australia, #231222), Sanger sequencing was performed at the AGRF to determine the sequence of *VviPLATZ1* in 04C023V0006, 04C023V0003 and 03C003V0060.

**mRNA in situ hybridization.** In situ hybridization of *VviPLATZ1* was performed in sectioned 04C023V0006 (*H/H*), 04C023V0003 (*f/f*) and 03C003V0060 (*M/f*) flowers at stage six WPA using a modified procedure previously described[66]. Briefly, inflorescences were fixed in 4% paraformaldehyde (PFA) phosphate buffered solution (pH 7.2) containing 0.5% Triton X-100 and 5% DMSO for 12 h at 4 °C. Floral tissues were dehydrated using a graded ethanol series (H₂O, 15%, 30%, 40%, 50%, 60%, 70%, 80%, 90%, 95% and 100%) at 4 °C and cleared using a graded Histoclear solution (Sigma-Aldrich, Sydney, New South Wales, Australia, #H2779-1L; 25%, 50%, 75% and 100%) at 20 °C before embedding tissue in Paraplast Plus (Sigma-Aldrich, Sydney, New South Wales, Australia, #125387-89-5). Microtome sections (8–10 µm thick) were prepared and adhered to ProbeOn Plus™ microscope slides (ThermoFisher Scientific, Thebarton, South Australia, South Australia, #15-188-51) followed by deparaffinization with Histoclear. Next, slides were rehydrated via a graded ethanol series (100%, 95%, 85%, 70%, 50%, 30%, 15%, H₂0) at 20 °C then treated with proteinase K (Promega Madison, Wisconsin, USA, #EO0491; 1.0 µg/ml in 100 mM Tris-HCl, pH 7.5 plus 50 mM EDTA) at 37 °C for 30 min. After this treatment, slides were fixed with 4% formaldehyde in phosphate-buffered solution (pH 7.2) at 20 °C. Next, slides were treated with 0.1 mM triethanolamine pH 8.0 (Sigma-Aldrich, Sydney, New South Wales, Australia, #90279-100 ML) for 10 min at 20 °C, then dehydrated in a graded ethanol series (H₂O, 15%, 30%, 50%, 70%, 85%, 95%, 100%) at 20 °C and dried under vacuum. A digoxigenen (DIG) labeled anti-sense probe from nucleotide position 54 to 870 in the *VviPLATZ1* coding sequence was produced using the T7 polymerase (Promega, Sydney, New South Wales, Australia, #P207). Hybridization was performed in saline-sodium citrate (SSC) solution containing 1% SDS, 50% formamide, 100 µg/mL tRNA for 12 h at 52 °C. After washing with SSC solution, the slides were incubated with the Boehringer Blocking Reagent (Roche Diagnostics, Mannheim, Germany, #11096176001) for 2 h at 4 °C. Next, the slides were incubated with a 1:1000 dilution of anti-DIG-alkaline phosphatase-conjugated Fab fragments (Roche Diagnostics, Mannheim, Germany, #11093274910), in phosphate-buffered solution (pH 7.2) containing 0.2% BSA and 0.1% Triton X-100 for 2 h at 4 °C. After washing with phosphate-buffered solution (pH 7.2), the slides were incubated with the alkaline phosphatase substrate Western Blue® (Promega, Sydney, New South Wales, Australia, #S3841) for

12 h at 20 °C in order to visualize of *VviPLATZ1* transcripts. Colorization was terminated by incubating slides in TE buffer (10 mM Tris-HCl/1 mM EDTA, pH 7.5). The sections were mounted with glycerol and imaged using an optical microscope (Zeiss Axioskop2, Oberkochen, Germany). Three biological replicas were derived from 04C023V0006, 04C023V0003 and 03C003V0060 plants. Each biological replicate was derived from a single inflorescence.

**CRISPR/Cas9 vector construction**. The *VviPLATZ1* DNA sequence was scanned for 20 bp guide sequences followed by the NGG protospacer adjacent motif (PAM) using Benchling (https://benchling.com), which also calculated on and off target scores[67,68]. The two guide RNA sequences, FS1 and FS4, that targeted exon 2 were selected, as gene-edited mutations would result in the production of truncated proteins lacking the conserved PLATZ domain. These guide RNAs were also selected for their high cleavage efficiency using the Guide-it sgRNA In Vitro Transcription and Screening kit (Takara Bio USA, Inc).

The *CRISPR-associated protein-9* nuclease gene (*Cas9*)-single guide RNA (sgRNA) cassette was synthesized at GenScript Biotech (Piscataway, NJ, USA) then subcloned into the pCLB1301 binary vector (Supplementary Fig. 9)[37]. The *Streptococcus pyrogenes Cas9* sequence containing nuclear localization sequences and the potato *IV2* intron[69] was codon optimized for *Vitis*. Cas9 transcription was controlled by the Arabidopsis *UBIQUITIN1* promoter and terminator[70], which drives expression in vegetative and reproductive tissues[71]. The *UBIQUITIN6* promoter was used to drive the expression of FS1 and FS4 sgRNAs[72]. The two binary vectors pVCAS9FS1 and pVCAS9FS4 contained the FS1 or FS4 guide RNA, respectively. Lastly, these binary vectors contained the hygromycin resistance gene (HygR), and the endoplasmic reticulum localized green fluorescent protein (GFP-5ER) reporter for selecting transgenic plants[73].

**Transformation and identification of gene-edited *VviPLATZ1* plants**. To edit *VviPLATZ1*, pVCAS9FS1, and pVCAS9FS4 were transformed into the *Agrobacterium tumefaciens* strain EHA105 and selected using the kanamycin resistance gene, KanR. Next, transgenic 04C023V0006 (*H/H*) plants were produced by incubating the EHA105 strains with somatic embryogenic callus derived from anthers[74]. Transgenic somatic embryos were selected using the HygR and GFP-5ER[75]. Shoot growth was induced in transgenic somatic embryos with 5 μM 6-Benzylaminopurine (Sigma-Aldrich, Sydney, New South Wales, Australia, #B327). After shoots developed, root growth was induced with 0.5 μM 1-naphthaleneacetic acid (Sigma-Aldrich, Sydney, New South Wales, Australia, #N0640). Finally, rooted transgenic plantlets maintained in tissue culture, were propagated, potted into soil and transferred to the glasshouse and/or growth chamber for further analysis.

To identify gene-edited plants, DNA isolation was performed on immature leaves from GFP-positive plantlets. The primers Pla_F (ATAAGGCTCAACCCCCACTT) and Pla_R (ACACCCCAATAAAACGCAAA) were used to PCR amplify the target regions for the guide RNAs in the second exon of *VviPLATZ1*. Amplicon Sanger sequencing followed by chromatogram analysis was used to identify indels in the GFP-positive plants. Using the next-generation sequencing service at AGRF, gene-edited *VviPLATZ1* were analyzed further using the MiSeq sequencing system (Illumina, San Diego, CA, USA). Subsequently, MiSeq sequencing data were processed through the CRISPResso pipeline to identify indels[76]. Mutation efficiency was calculated by counting the number of sequence reads with an indel at the guide RNA target region divided by the total number of sequence reads. The value was corrected by subtracting the data obtained for the untransformed 04C023V0006 (*H/H*) control. After this analysis was completed, seven first-generation plants (T$_0$) containing gene-edited alleles were identified and self-pollination was performed to evaluate the flower sex phenotype in T$_1$ plants. After DNA isolation from T$_1$ root and cotyledon tissues, the guide RNA target region was amplified by PCR using Pla_F and Pla_R primers. Sanger sequencing performed at AGRF was used to genotype the embryos to identify T$_1$ plants homozygous for the mutated alleles (*Vviplatz1-1/2-8*).

**Flower sex determination**. Phenotyping for flower sex was performed by morphological scoring using the OIV descriptor, No 151, of recently opened flowers (https://www.oiv.int/public/medias/2274/code-2e-edition-finale.pdf). The architecture of each stamen was scored as (1) elongated and upright, (2) reflexed or (3) stunted and upright. A minimum of 30 flowers from at least 2 inflorescences were scored from each genotype. Images of open flowers and stamens were taken with a Spot 15.2 64 Mp camera mounted on a Zeiss Stemi 2000-C dissecting microscope (Zeiss Vision Australia Pty Ltd, Tonsley, SA, Australia) using the Spot V5.1 software (SPOT Imaging, Sterling Heights, MI, USA). Stamen length was measured in flower images from 04C023V0006 (*H/H*), 04C023V0003 (*f/f*) and T$_1$ homozygous plants using the ImageJ.app. (https://imagej.net/Fiji/Downloads). Analysis of variance and Tukey's honest significant difference analysis was performed using standard statistical packages in R.

**Statistics and reproducibility**. Two-tailed Student's *t* test was used to determine statistical differences with *p* values provided for all comparisons. This information is provided in figure legends with raw data provided in the Source Data file. Measurements were taken from multiple biological samples. For in situ

hybridization, three independent inflorescences were fixed, processed, sectioned, and hybridized in separate experiments with similar results produced and a representative image displayed in Fig. 2.

**Reporting summary**. Further information on research design is available in the Nature Research Reporting Summary linked to this article.

## Data availability

All relevant data presented in this manuscript are available within the article and its Supplementary Information files. The GenBank sequence identifier for *VviPLATZ1* derived from 04C023V0003 is MW548436. For *Vviplatz1* derived from 04C023V0003, the GenBank sequence identifier is MW548435. Source data are provided with this paper.

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

## Acknowledgements

The authors thank Debra McDavid for technical assistance with the transformation, Helen Le for maintaining tissue culture plantlets, Jiao Li for technical assistance with gene-editing and Rosanna Powell for help with the ImagJ.app. The authors thank Ian Dry, Steve Henderson and Jake Dunlevy for helpful discussions with this manuscript, Christine Böttcher and Crista Burbidge for assistance with RT-qPCR and Paul Thomas from the University of Adelaide/SAHMRI for useful feedback on CRISPR/Cas9 vector design and gene-editing. This work was funded by Wine Australia (CSP 06/01) and CSIRO Agriculture and Food.

## Author contributions

M.R.T. and L.T. designed and supervised the genetic mapping; M.R.T. designed and supervised the CRISPR/Cas9 gene-editing work. J.C. identified the SEX LOCUS SNPs, performed the genetic mapping and identified *VviPLATZ1* as a candidate gene. P.C. performed the RT-qPCR analysis, cloning, CRISPR/Cas9 gene-editing and phenotyping. D.M. participated in the flower sex phenotyping for mapping, breeding and maintenance of plants in glasshouse and growth cabinets. H.M.S. performed the mRNA in situ hybridization and wrote the manuscript with input from the authors.

## Competing interests

The authors declare no competing interests.
