## [Peer Review File · Nature Communications]

VviPLATZ1 is a major factor that controls female flower morphology determination in grapevineReviewers' Comments:

Reviewer #1:

Remarks to the Author:

The article provides new important insights into the role of a transcription factor in grapevine flower morphology. However, we found major issues that need to be addressed before the article can be accepted for publication.

Grape species are dioecious except the cultivated grapevine, *Vitis vinifera* ssp. *vinifera*, which is mostly hermaphrodite. While male and hermaphrodite flowers have erect stamens with fertile pollen, female flowers generally possess reflexed stamens that carry few sterile pollen grains. In this study, the authors show that the loss of function of the transcription factor VviPLATZ1 leads to flowers with reflexed filaments, and therefore propose VviPLATZ1 as the major determinant for flower sex determination in *Vitis*. Using SNP genotyping of 19 SNPs spanning the SEX LOCUS (SL), the authors describe 15 SNPs associated with flower sex type in the F1 population 00C001V0008 (f/f) x Ugni Blanc (H/f), suggesting that the SL starts in the TREHALOSE-PHOSPHATE PHOSPHATASE gene (TPP). Sequencing of VviPLATZ1 from 04C023V0006 (H/H) and 04C023V0003 (f/f) allowed to identify 10 SNPs that may affect the transcription and/or the function of the protein. In addition, the transcript level for VviPLATZ1 was found significantly higher in inflorescences of 04C023V0006 (H/H) compared to 04C023V0003 (f/f) at early development. In situ hybridization of VviPLATZ1 mRNA revealed a low expression of the gene in the filaments and anthers of female flowers. Edition of VviPLATZ1 in a hermaphrodite genotype (H/H) using CRISPR/Cas9 system gave rise to individuals carrying flowers with reflexed stamens.

Major reviews:

1. The phenotypes of the VviplatZ1 mutants suggest that VviPLATZ1 transcription factor is involved in the morphology of the grapevine flower stamens. While reflexed filaments likely reduce self-pollination in female flowers, the lack of pollen tube germination is the major factor of male sterility. Accordingly, while the characterization of VviPLAZ1 is an important step in the dissection of the sex locus the authors did not demonstrate the role of the gene in sex determination; we therefore strongly recommend that the authors revise the claim that VviPLAZ1 is a major determinant for flower sex determination and focus on its role on stamen morphology.
2. The SNP genotyping analysis does not provide any new findings on the sex-determining region, nor the comparison of VviPLATZ1 gene sequence and the differential gene expression of VviPLATZ1 in developing inflorescences. These results should be presented in light of the extensive body of information in the literature. In addition, the set of SNPs used in the study does not cover the entire grape sex-determining region, which starts at the 5' of the gene VviYABBY3 to downstream of the 3' terminus of VviAPT3 (Massonnet et al., 2020; Zou et al., 2021 PNAS <https://doi.org/10.1073/pnas.2023548118>); t
3. The genome used as reference is not appropriate. The genome of PN40024 is not appropriate as reference for detecting SNPs associated with flower sex trait, because the sex-determining region on chr2 is a mixture of H and f haplotypes (see Zhou et al., 2019 Nature Plants). We suggest using a H or f haplotype of the region as reference for the SNP analysis.
4. Only two sibling individuals, one hermaphrodite and one female, were used for gene expression evaluation, in situ hybridization of VviPLATZ1 mRNA, and VviPLATZ1 sequence comparison. All these assays should include several individuals with different genetic background per flower sex type, including male type, to be able to conclude that the polymorphisms found, including the ones in VviPLATZ1 sequence, as well as VviPLATZ1 expression patterns, are associated with stamen erectness in *Vitis* sp.

5. The low expression of VviPLATZ1 in stamens of female flowers and the phenotype of the vviplat1 mutants constitute new findings. As mentioned above, a comparison of gene expression in flower tissues should include several individuals carrying flowers with erect stamens, including male and hermaphrodite individuals.

6. It has been hypothesized that reflexed filaments in female flowers are due to the smaller size of the epidermal cells on the abaxial (dorsal) side compared to the adaxial side (Dorsey, 1914). Thus, it would be relevant to indicate if differences in filament tissue were observed between hermaphrodite and female flowers at the stage used for in situ hybridization of VviPLATZ1 mRNA, i.e. 6 weeks before anthesis.

7. Finally, claims on the impact of VviPLAZ1 on pollen viability should be supported by experiments that test pollen germination of the vviplat1 mutants compared to the control 04C023V0006 (H/H),

Detailed comments

Title

- Page 1, line 1: "Grapevine flower sex determination is controlled by the VviPLATZ1 transcription factor". The results presented in the study, particularly the phenotype of the vviplat1 mutants, suggest a role in the straightness/erectness of the grapevine flower stamens. However, results do not demonstrate the role of VviPLATZ1 in male fertility as claimed repeatedly in the manuscript. Therefore, the title is catchy, but misleading, and needs to be modified.

Introduction

- Page 2, line 52: "While pollen derived from female flowers is viable, fertility is reduced due to an impairment in germination¹⁷". In their study, Caporali et al. (2002) mentioned that female flowers produce sterile pollen. "Sterile microspores and pollen grains in female flowers display an abnormal round shape, lacking colpi and possessing uniformly thickened cell walls that impede germination." We suggest revising the sentence to explain that male sterility in female flowers is due to the absence of pollen germination, not a reduction.

- Page 2, line 57: "SEX LOCUS (SL)". We would suggest naming this region "sex-determining region" in accordance with the recently published studies (Massonnet et al., 2020; Badouin et al., 2020; Zou et al., 2021).

- Page 3, line 78: "As VviYABBY3 is located outside and adjacent to the SL, it is unclear if this gene is the primary M-factor that specifies male flower identity²⁶". This is another statement not supported by the literature. VviYABBY3 is located inside the sex-determining region according to two recent studies (Massonnet et al., 2020; Zou et al., 2021).

Results

- Page 3, line 94: The genome of PN40024 is not appropriate as reference for detecting SNPs associated with flower sex trait, because the sex-determining region on chr2 is a mixture of H and f haplotypes. We would suggest using a H or f haplotype of the region as reference for the SNP analysis.

- Page 3, line 107: "Together, these results are consistent with genetic studies showing that the 5' boundary of the SL is located in TPP25,26." In absence of male individuals in the study, the identified region cannot correspond to the entire sex-determining region. This is a critical flaw of the study.

- Page 3, line 115: "Transcript levels for VviPLATZ1 was examined in 04C023V0006 (H/H) and compared to 04C023V0003 (f/f) at -8, -6 and -4 stages of flower development previously defined³⁵." To help the reader, we would suggest explaining that -6 stage corresponds to flowers at 6 weeks post-anthesis, for instance. Pictures of the inflorescences at the collection time would also help.

- Page 4, line 127-134: We suggest comparing the mutation events identified in this study with the ones described in studies recently published (Massonnet et al., 2020; Badouin et al., 2020; Zou et al., 2021).
- Page 4, line 129: "First, a T to A point mutation and adenine insertion were identified in the first codon, resulting in a frame-shift mutation that altered the position of the start codon (Fig. 1d)." The described mutations would shorten the f protein of 2 amino acids. What would be the impact of this shortening on the protein function?
- Page 4, line 135-136: "As the phenotype is recessive, the female allele is referred to as Vviplat1." Lower case is commonly used for naming unfunctional allele (as used by the authors for naming knockout mutants). No deleterious mutations were identified in the f allele of VviPLATZ1 and the f version of the protein VviPLATZ1 was not showed to be unfunctional. We would suggest naming VviPLATZ1 alleles according to their sex-determining haplotypes. i.e. f allele of VviPLATZ1 and H allele of VviPLATZ1.
- Page 4, line 138-141: "To gain insight into the proposed function of VviPLATZ1 in stamen development, the expression pattern of this gene was analyzed in 04C023V0006 (H/H) and 04C023V0003 (f/f) flowers at stage -6 of flower development." Which criteria were used to select the -6 developmental stage for in situ hybridization of VviPLATZ1 mRNA? Were differences between female and hermaphrodite filaments observed?
- Page 4, line 142-143: "A similar pattern of expression was detected in male flowers (data not shown)." Not showing these data is not acceptable. Data from male individuals need to be shown to demonstrate the difference of expression between erect and reflexed filaments. In addition, we suggest including several individuals with different genetic background per sex type to show the low expression of VviPLATZ1 in stamen tissues of female flowers across the Vitis genus.
- Page 4, line 168-171: "The average length of stamens in hermaphrodite individuals was 3.0 mm. In contrast, female flowers initiated reflexed stamens 98% of the time (Fig. 4b, g). The remaining 2% of female stamens displayed an upright growth pattern but were significantly shorter with an average length of 1.2 mm." Data are not provided, nor the statistical significance. How many flowers per plant were measured?
- Page 4, line 173: "The morphology of Vviplat1-1/2, -4, and -6 were also similar to female flowers (data not shown)." We are not sure why authors don't provide this critical information that can be easily added, if not to the manuscript, at least to supplementary files.

Methods

- Page 6, line 240: Methods of SNP genotyping needs to be provided.
- Page 6, line 251: Authors should indicate if biological replicates were used for evaluating VviPLATZ1 expression.

Reviewer #2:

Remarks to the Author:

This ms by Iocco-Corena et al. is a contribution to identifying the sex-determining genes in grapevine. Sex determination is much less known in plants than in animals, but these 5-6 years great advances have been made in finding plant sex-determining genes. Two papers on a genomic characterization of the grapevine sex locus came out last year and several good candidate sex-determining genes were pointed out (Badouin et al. 2020, Massonnet et al. 2020). No functional validation however was

carried out. This ms reports a Crispr-cas9 validation of the VviPLTAZ1 gene using microvines. They showed that this gene underlies reflex stamens in females, an important aspect of floral morphology in grapevine that was left unexplained in previous work (Badouin et al. 2020, Massonnet et al. 2020). I found the use of microvines and Crisp-cas9 very elegant. This is an important contribution to understanding grapevine sex determination. It should be noted that functional validation of sex-determining genes is rarely done in plants since not all plants can be easily transformed, which makes this contribution also important for the whole field of plant sex determination. I am thus very positive about this work. I however have several comments:

1) The introduction of the ms includes a discussion of the papers I mentioned above (Badouin et al. 2020; Massonnet et al. 2020). I did not think that these papers were very well presented. Both papers agreed on VviINP1 being a male-determining gene. They however differed for the female-determining one. Badouin et al. suggested VviAPT3 was the female-determining gene, Massonnet et al. suggested VviYABBY3 was the one. I think this part of the introduction should be re-written.

2) Previous work disagreed about the boundary of the sex locus. Badouin et al. considered VviYABBY3 outside of the sex locus, while Massonnet et al. considered it inside. Revisiting this question is thus helpful. Iocco-Corena et al. found that both VviAPT3 and VviYABBY3 are outside of the sex locus. However, I think that their results should be interpreted more cautiously. First, they have used the 12X version of PN40024, in which the sex locus is not well assembled (VviAPT3 is missing). This might have affected how they located their markers in the sex locus. This should be discussed. Second, they found a recombination event at the VviMT72 marker, but recombination between f and H haplotypes are known to have happened and some H haplotypes have in fact the f version of VviAPT3. Their results might hold only for cultivated grapevines and not wild ones. Unless, they can show evidence that VviAPT3 is not genetically linked to the sex locus in wild grapevine, they should make it clear that their interpretation is restricted to cultivated grapevines (and might even not be valid for all cultivated grapevines). This is important as in previous work (in particular in Badouin et al.) the boundaries of the sex locus have been studied in wild grapevines. It might be confusing for the readers if this point is not clarified and discussed. Also, it means that they cannot really rule out VviAPT3 as being a sex-determining gene (see also my comments 4 and 5).

3) The mutations that they found in VviPLTAZ1 (f version, see Figure 2) are very interesting and might explain why its expression is reduced in females. Looking at male individual (of *V. sylvestris*) would have been very interesting to verify that these mutations are absent in the M haplotype.

4) Their expression study of VviPLATZ1 is interesting and does provide useful data compared to previous work. However, I am not convinced by the interpretation about the expression of this gene in the ovule. They say: « Interestingly, VviPLATZ1 was expressed in the ovule of hermaphrodite and female flowers (Fig. 2a, d, respectively). The molecular studies above together with the reduced expression of VviPLATZ1 in developing stamens of female flowers indicates that this gene is involved in flower sex determination. » and in the abstract they say : « Taken together, the results demonstrate that loss of VviPLATZ1 function is THE major determinant for flower sex determination in *Vitis*. » I think they have shown that VviPLATZ1 is an important determinant of flower sex determination, I don't think that they have shown it is THE major determinant. The role of VviPLATZ1 mutation in producing reflex stamens in female flower is clear, the role in preventing ovule production is not clear at all. The above sentences seems to suggest that there is only one sex-determining gene in grapevine with both male and female functions. Classical genetic experiments suggested that sex is determined by two genes in grapevine. I don't think the authors have gathered evidence that this is not true and VviPLATZ1 is the only determinant (see also next comment). This should be clearly explained in the ms.

5) Another important point is about the manipulation of VviPLATZ1 using Crispr-cas9. The results are very neat, they do show that VviPLATZ1 is underlying the reflex stamens phenotype. However, differences in flower morpho-anatomy between male and female are more than just reflex stamens. Pollen is infertile in female flowers. This has not been looked at in the Crispr-cas9 modified plants. There is no evidence that VviPLATZ1 is underlying this floral trait as well.

Minor comments :

I think a few words are needed to explain what are the microvines, first time they are mentioned in

the text, for the readers who are not grapevine experts.

Reviewer #3:

Remarks to the Author:

Manuscript background information

All grapevine species are dioecious, with the only exception being the cultivated variety, *V. v. vinifera*. Hermaphroditism in cultivated *V. v. vinifera* is a trait that was gained during domestication. In this paper the authors, Iocco-Corena and co-workers, tried to go deeper in a century old question: what makes grapevines dioecious? The authors analysed the female sex locus region and identified the *VviPLATZ1* transcription factor as a candidate flower sex factor. The authors used the CRISPR/Cas9 system to edit *VviPLATZ1* (loss of function) in a hermaphrodite genotype and showed that individual lines homozygous for the gene edited alleles gave rise to female flowers with reflex stamens. An evidence of *VviPLATZ1* is involved in flower sex determination in *Vitis*.

Comments for transmission to the authors

In general, this is an interesting research paper. However, it requires a minor revision of the English language, since in few instances the manuscript lacks fluidity possible due to several authors contributing. The discussion is very well based on results and the methodology seems robust. There are, however, some small questions and remarks.

1 – I do not think references in the abstract is a common occurrence. I would advise removing it from the abstract.

2 – The names of the genes are in italics. Please review the article.

3 – In Line 83-85 It was not clear why, among several genes in the SEX LOCUS, the authors decided to focus on *VviPLATZ*. Could you explain the reason, please? You state that through genetic mapping you identified *VviPLATZ1* but no reference is supplied.

4 – Authors made very clear the role of *VviPLATZ* in the stamen formation in female grapevine. However, in the plants bearing the edited alleles of *PLATZ* it would be interesting to assess the expression value of other putative important genes in the female portion of the sex locus, namely *VviINP1* and even *VviAPRT3*, *VviYABBY* and *VviFSEX*. Although I understand that this might not be in the scope of the manuscript.

Line 15

"modern domesticated *Vitis vinifera* cultivars form hermaphrodite flowers capable of self-pollination" please bear in mind that some female cultivars of *V. vinifera* exist for wine production making this phrase not entirely true.

Line 18

"To identify the recessive allele responsible for the development of reflex stamens" I do not agree with the use of the word "recessive" in the abstract, merely because the authors haven't yet explained why it should be a recessive allele. Although scientists familiarized with the *Vitis* sex dilemma recognize this fact other researchers may not have this perception.

Also, in the abstract the authors claim that *VviPLATZ* is the major player for flower sex determination (Line 27). The use of the term "major" implies knowledge of the function of the other genes in the sex

region. At the moment, this knowledge is not yet available. Therefore, I recommend some caution when using this term.

Line 28-29

"Traits associated with flower development and fertilization are key for increasing yield in agriculture and horticulture crops" – either a reference is missing or please replace "are" for "could be".

Line 33

"...underlying mechanisms that control flower identity are distinct." – A reference is missing at the end of this sentence.

Line 46-48

Please bear in mind that several female cultivars of *V. vinifera* exist and are used for wine production. This means that, *V. vinifera* cultivars for wine are not exclusively hermaphrodites.

Line 80

I think it is important for the authors to refer to the putative importance of *VviINP1* and the 8bp deletion potentially involved in pollen infertility in female plants and *VviFSEX* which, although some groups show a non-specific location of mRNA and non-differentially expressed (Coito et al., 2017) other groups showed to be differentially expressed (Zhou et al 2017, Badouin et al., 2020).

Line 98

Out of curiosity could you state the germination rate of the crosses?

Line 103-104

"genotyping self-fertilized hermaphrodite microvine (H/f) progenies (S1 to S5)", could you provide the segregation rate? According to my calculation it should result in a 3:1 hermaphrodite to female rate (or a 2:1:1 of h/f, h/h and f/f).

Line 104-105

Out of curiosity, how does the SNP set used link with *VviAPRT3* and *VviFSEX* from Coito et al., 2017 and *VviINP1* of Badouin et al., 2020 and Massonnet et al., 2020?

Line 109-111

Before you stated that the SL contains 11 genes. However, you narrow the SL to only seven genes. Are these seven genes responsible for the female identity? Please clarify. How does this compare with the recent works of Badouin et al., 2020 and Massonnet et al., 2020?

Line 115

Although referenced, I think the manuscript would benefit greatly with representative images of the development stages used: -8, -6 and -4.

Line 124-125

I think a correction should take place in this sentence. Badouin et al., 2020 reports a higher expression of *VviINP1* in female flowers.

Line 127 to 132

The *VviPLATZ* gene that the authors refer to is not the one located on chromosome2 in the database referred to in Fig1 (grapegenomics.com). This gene with the described sequence is located on the unknown chromosome. I would suggest that the authors contextualize these locations and why you chose to work with the unknown chromosome allele

Line 144-145

Why do the authors consider this observation interesting? What about male plants? Do they also show

VviPLATZ mRNA in the ovule? If only hermaphrodites and female plants show this presence, then VviPLATZ might play a more complex role in flower formation. From my understanding male plants still display ovules.

Line 159-161

Despite showing altered stamens all plants bearing the altered PLATZ could still produce viable pollen? Sometimes pollen from wild female plants is viable and produces offspring.

Line 170-171

Grapevine stamens polymorphism was described previously and so, despite being old, would be nice if could cite the work of Stout, 1921 (<http://www.mavo.biz/Reprints/Stout1921.pdf>) where polymorphism of grapevine stamens is described.

Line 195

What is the relation of maize flouy3 and rice GL6 with VviPLATZ family? Are they homologues, orthologues? Please clarify.

Line 210-211

"As a result, gene editing of VviINP1 is necessary to validate a role for the 8-bp deletion in pollen fertility." Just a minor note, but I think the phrase would be better as "As a result, gene editing of VviINP1, in grapevine, is necessary to validate a role for the 8-bp deletion in pollen fertility."

Line 211-213

"However, VviPLATZ1 may also control pollen fertility given its expression in microspores (Fig. 2c, f), close proximity to VviINP1 within the SL (Fig. 1a, b)

and role as a transcription factor." This sentence is not very clear.

But if understand correctly, you self pollinate plants with the edited VviPLATZ. So the pollen was fertile? Was the number of fruits produced in this self-pollination different from a self-pollination of non edited plants?

Perhaps the photos shown in figure 2 are not the best to demonstrate the expression of VviPLATZ in the microspores. The signal is weak in the female plant and the hermaphrodite has some background noise. Perhaps in this case it would be advisable to place the images of the controls even in annex.

Line 226

This section of the methods is a bit confusing. As I understand, F1 plants and S1 to S5 were already established from previous studies (reference 34 and 47) however the way the manuscript is written made me think that you edited VviPLATZ and then self-pollinate the edited grapevine for five generation. I think this should be clarified.

Line 228-229

"00C001V0008 (f/f) x Ugni Blanc (H/f) individuals and successive self-pollinated microvines229 (H/f) from first (S1) to fifth generation (S5) as previously described"

The individual 00C001V0008 is a microvine? Is also derived from the Grenache (H/f) x L1 microvine cross? Also how many time between generations elapsed? Were the plants supplied with hormones in order to accelerate the life cycle and produce flowers early?

This article was reviewed by Margarida Rocheta and João Lucas Coito from Instituto Superior de Agronomia, Lisbon, Portugal

Reviewer #4:

Remarks to the Author:

All of the approximately 70 species in the genus *Vitis* are dioecious, and only the cultivated grapevine, *Vitis vinifera* subsp. *vinifera*, has reverted to hermaphroditism, while its wild ancestor, *V. vinifera* subsp. *sylvestris*, retains the dioecious condition. The shift occurred during domestication about 8,000 years ago. Pat Iocco-Corena and colleagues have used genetic mapping and expression studies to identify a candidate transcription factor that regulates stamen architecture during grapevine flower development, which they have christened VviPLATZ1 (plant AT-rich sequence-and zinc-binding protein1). They used the CRISPR/Cas9 system to edit VviPLATZ1 in a hermaphrodite genotype (of cultivated wine). Phenotype evaluation showed that individuals with gene-edited alleles of VviPLATZ1 had flowers with reflexed architecture, that is, flowers that are functionally female because they cannot self-pollinate. VviPLATZ1 thus is a novel regulator of stamen architecture.

The results are clear, and the study is a great step forward. As the authors point out, a candidate gene suggested to control pistil suppression in male flowers of wild wine, namely VviAPT3, and another involved in inaperturate pollen (VviINP1), causing male-sterility candidate in wild wine, still need to be tested with the CRISPR/Cas9 system.

There are numerous sentences in which the words 'gene edited' need to be hyphenated: gene-edited VviPLATZ1 alleles; gene-edited mutations; gene-edited VviPLATZ1 plants. gene-edited VviPLATZ1 – plants missing; etc.

Also, search for adn instead of and

RESPONSE TO REVIEWERS

All references to figures are highlighted in yellow

All references to lines in the revised manuscript are highlighted in blue

Reviewer #1 (Remarks to the Author):

The article provides new important insights into the role of a transcription factor in grapevine flower morphology. However, we found major issues that need to be addressed before the article can be accepted for publication.

Grape species are dioecious except the cultivated grapevine, *Vitis vinifera* ssp. *vinifera*, which is mostly hermaphrodite. While male and hermaphrodite flowers have erect stamens with fertile pollen, female flowers generally possess reflexed stamens that carry few sterile pollen grains. In this study, the authors show that the loss of function of the transcription factor VviPLATZ1 leads to flowers with reflexed filaments, and therefore propose VviPLATZ1 as the major determinant for flower sex determination in *Vitis*. Using SNP genotyping of 19 SNPs spanning the SEX LOCUS (SL), the authors describe 15 SNPs associated with flower sex type in the F1 population 00C001V0008 (f/f) x Ugni Blanc (H/f), suggesting that the SL starts in the TREHALOSE-PHOSPHATE PHOSPHATASE gene (TPP). Sequencing of VviPLATZ1 from 04C023V0006 (H/H) and 04C023V0003 (f/f) allowed to identify 10 SNPs that may affect the transcription and/or the function of the protein. In addition, the transcript level for VviPLATZ1 was found significantly higher in inflorescences of 04C023V0006 (H/H) compared to 04C023V0003 (f/f) at early development. In situ hybridization of VviPLATZ1 mRNA revealed a low expression of the gene in the filaments and anthers of female flowers. Edition of VviPLATZ1 in a hermaphrodite genotype (H/H) using CRISPR/Cas9 system gave rise to individuals carrying flowers with reflexed stamens.

Major reviews:

1. The phenotypes of the VviplatZ1 mutants suggest that VviPLATZ1 transcription factor is involved in the morphology of the grapevine flower stamens. While reflexed filaments likely reduce self-pollination in female flowers, the lack of pollen tube germination is the major factor of male sterility. Accordingly, while the characterization of VviPLAZ1 is an important step in the dissection of the sex locus the authors did not demonstrate the role of the gene in sex determination; we therefore strongly recommend that the authors revise the claim that VviPLAZ1 is a major determinant for flower sex determination and focus on its role on stamen morphology.

Response:

To address this comment by reviewers #1, the title of the manuscript has been modified to "A major determinant of grapevine flower sex identity is controlled by the VviPLATZ1 transcription factor". Further, we have maintained focus on the role of VviPLATZ1 in female flower identity and morphology, as the development of reflex stamens is the major morphological phenotype of female flower morphology.

2. The SNP genotyping analysis does not provide any new findings on the sex-determining

region, nor the comparison of VviPLATZ1 gene sequence and the differential gene expression of VviPLATZ1 in developing inflorescences. These results should be presented in light of the extensive body of information in the literature. In addition, the set of SNPs used in the study does not cover the entire grape sex-determining region, which starts at the 5' of the gene VviYABBY3 to downstream of the 3' terminus of VviAPT3 (Massonnet et al., 2020; Zou et al., 2021 PNAS <https://doi.org/10.1073/pnas.2023548118>); t

Response:

We politely disagree with reviewers #1 that the mapping doesn't provide new information.

1. The mapping data presented in this manuscript was completed in 2010, well before the sequence analyses were performed by (Zhou et al., 2019; Badioun et al., 2020; Massonnet et al., 2020; Zou et al., 2021). Instead of publishing our results, we worked to functionally validate the VviPLATZ1 female candidate factor.
2. Massonnet et al., 2020 genetically dissected the SDR into two regions. Results from the association analysis indicates that the male flower identity region is located in the 5' region of the SDR from VviYABBY3 to the aldolase gene. They also provide evidence that female flower identity is specified by a genetic determinant downstream of the aldolase gene and includes sex linked genes from TPP to VviAPT3 (12 genes total). While our manuscript was in review, the Zou et al., 2021 manuscript was published. Results from this manuscript narrowed the male identity to include VviYABBY3 and VviSKU5. In addition, VviAPT3 was excluded from the female identity region indicating that this region is composed of 11 genes from TPP to WKRY21. Based on the results of Massonnet et al., 2020 and Zou et al., 2021, we have updated Fig. 1a, which displays the SDR of the Cabernet Sauvignon *f*-haplotype, as suggested by reviewers #1. In this image, we highlighted the genes in the female identity region with yellow arrows and used grey arrows to mark the other genes that localize to the SDR. In our manuscript, we identified 19 SNPs spanning the female flower identity region from the aldolase gene to LPPB, downstream of VviAPT3. After genotyping the F1 00C001V0008 x Ugni Blanc individuals and selected domesticated grapevine cultivars, we showed that 5' boundary of the female identity region is located in the TPP gene (Fig. 1b). Further, we genotyped self-pollinated (S1 to S5) microvines segregating for female and hermaphrodite individuals and results showed that the 3' boundary of the female identity region is located in FMO4 (Fig. 1b). Thus, the recombination events in TPP and FMO4 are significant, as the fine genetic mapping narrowed the number of genes in female identity region to seven, from VviINP1 to FMO3. Therefore, our fine genetic mapping excludes TPP, FMO4, VviSEX and WRKY21 as candidate genes implicated in female flower sex determination. We have modified the text in results section to increase the clarity and significance of our genetic mapping results.

“Genetic and molecular analysis of VviPLATZ1. As shown in Fig. 1a, association genetics indicates that the female identity region is comprised of eleven genes from *TPP* to *WRKY21*³¹. To further narrow down the number of genes that reside in the female identity region, fine genetic mapping was performed using 21 SNPs spanning the SDR (Supplementary Table 1-3). One-hundred and one F1 00C001V0008 (*f/f*) x Ugni Blanc (*H/f*) individuals, which segregated for the formation of hermaphrodite and female flowers with a 1:1 ratio³⁷, were genotyped with the *SDR* SNP set. Results

showed that the 5' boundary of the *SDR* that specifies female flower identity was located in *TPP*, as indicated by the *VvMT_54* SNP (Fig. 1b; Supplementary Table 1, 2). This result was supported by genotyping 33 *Vitis vinifera* cultivars with known flower sex genotypes (*f/f*, *H/f* or *H/H*), using this SNP set. Self-fertilized hermaphrodite microvine (*H/f*) progenies (S1 to S5) segregating for hermaphrodite and female individuals with a 3:1 ratio were also genotyped with the *SDR* SNP set. A single recombination event at *VvMT72* in the S5-microvine, 06C008V0018, defined the 3' female flower identity boundary located in *FLAVIN-CONTAINING MONOOXYGENASE 4 (FMO4)*; Fig 1b; Supplementary Table 1, 2). Taken together, fine genetic mapping demonstrated that the female flower identity region in the *SDR* contained seven genes: *VviINP1*, *Exostosin*, 3-ketoacyl-acyl carrier protein synthase III (*KASIII*), *VviPLATZ1*, *FMO1*, *FMO2* and *FMO3* (Fig. 1b)." lines 157-243

3. Based on genetic mapping and gene expression data, we proposed that *VviPLATZ1* is a new candidate sex factor for female flower identity (lines 18-20; 125-128; 246-249). This is the first time *VviPLATZ1* has been implicated in specifying female flower identity. In addition, while sequencing studies have identified hermaphrodite, female and male *VviPLATZ1* alleles, none of the *SDR* publications have displayed DNA polymorphisms in the *VviPLATZ1* gene in light of its role in specifying female flower identity.
4. The gene expression studies presented in this manuscript provide new information and support the hypothesis that *VviPLATZ1* is a new candidate female sex identity factor. First, our expression studies are presented in light of stamen development rather than flower development. Second, we performed tissue specific expression studies to further support a role for *VviPLATZ1* in controlling stamen development. To date, tissue/cell type expression for the candidate female and male sex genes, *VviINP1* and *VviYABBY3*, as well as *VviAPT3*, have yet to be determined.
5. To address the reviewers #1 comment about the 19 SNPs not spanning the entire *SDR*, we have included two additional SNPs in the updated version of the manuscript that are located upstream of the aldolase gene. As results from Zou et al., 2021 indicate that the female identity region is contained in a region from *TPP* to *WRKY21*, the addition of these SNPs does not change the results derived from our genetic mapping.

Taken together, our fine genetic mapping results narrowed the female identity region from 11 to 7 genes. Through molecular and gene expression studies, we provide evidence that *VviPLATZ1* is a new candidate female sex identity factor that controls female flower formation.

1. The genome used as reference is not appropriate. The genome of PN40024 is not appropriate as reference for detecting SNPs associated with flower sex trait, because the sex-determining region on chr2 is a mixture of H and f haplotypes (see Zhou et al., 2019 Nature Plants). We suggest using a H or f haplotype of the region as reference for the SNP analysis.

1. As stated above, the mapping data presented in this manuscript was completed in 2010, well before the sequence analyses were performed by (Zhou et al., 2019; Badioun et al., 2020; Massonet et al., 2020; Zou et al., 2021). After performing the gene expression analyses, we worked to functionally validate the *VviPLATZ1* female candidate factor.
2. The suggestion by reviewers #1 is valuable; therefore, we have used the Cabernet Sauvignon *f* and *H* haplotypes as reference for the SNP analysis from grapegenomics.com. This information is presented in supplementary table 1 and 2. In addition, we used the Cabernet Sauvignon *f*-haplotype structure to display the results for the fine genetic mapping in Fig. 1a, b. However, in light of Nature's data policy, we are concerned about long-term data availability. NCBI is a permanent database system for managing genomic data. While grapegenomics.com is an extremely useful website for the grape community, the authors wonder about the long-term viability of the website. Previous grapevine genomic databases including VitisExpDB and <https://genomics.research.iasma.it> are defunct and <https://genomes.cribi.unipd.it> is no longer accessible as the connection is not private. Therefore, we maintained the SNP information for PN40024 in supplementary table 2.

4. Only two sibling individuals, one hermaphrodite and one female, were used for gene expression evaluation, in situ hybridization of *VviPLATZ1* mRNA, and *VviPLATZ1* sequence comparison. All these assays should include several individuals with different genetic background per flower sex type, including male type, to be able to conclude that the polymorphisms found, including the ones in *VviPLATZ1* sequence, as well as *VviPLATZ1* expression patterns, are associated with stamen erectness in *Vitis* sp.

Response:

1. The purpose of this study was to functionally validate the role for the loss of *VviPLATZ1* function in the formation of female flowers. Therefore, the molecular and expression studies were focused on genotypes used in the functional studies.
2. We have addressed reviewers #1 comment regarding *VviPLATZ1* sequence comparison from several individuals with different genetic backgrounds. Results in supplementary figure 2 show that the 3 SNPs at -13, -5 and +2 in the 5' UTR and start codon of 0C023V0003 (*f/f*) DNA are conserved in female *VviPLATZ1* alleles from a number of sequenced domesticated and wild *Vitis* species. The adenine insertion at +3 in the 04C023V0003 *VviPLATZ1* allele was found only in a subset of female *VviPLATZ1* alleles from domesticated cultivars. In supplementary figure 3, we show that the seven nonsynonymous SNPs identified in the coding region of 0C023V0003 (*f/f*) were also conserved in female *VviPLATZ1* alleles from a number of domesticated and wild *Vitis* species. Therefore, the 3 SNPs at -13, -5 and +2, as well as the nonsynonymous SNPs in the coding region of the 0C023V0003 (*f/f*) *VviPLATZ1* allele are conserved and therefore may play a role in reduced expression and/or function of this gene. Please refer to lines 265-282 in the results section.
3. Reviewers #1 suggested that we need to perform gene expression in several individuals with different genetic background per flower sex type. This is a surprising

statement given the recent publications in high impact journals including Nature Plants. Gene expression studies displayed in Zhou et al., 2019 and Badouin et al., 2020 were derived from RNA sequencing results performed by Ramos et al., 2014, which compared transcriptomic data between male, female and hermaphrodite flowers at four stages of development (note: a single male (wild *V. vinifera* accession not specified), female (a wild *V. vinifera* accession not specified) and hermaphrodite genotype (*V. vinifera* cv Touriga Nacional) was used in the Ramos et al., 2014 study. The authors of Zhou et al., 2019 and Badouin et al., 2020 displayed transcript profiles for the genes that localize to the SDR in their respective manuscripts. In the Massonnet et al., 2020 study, the gene expression data for SDR genes was derived from a hermaphrodite (*V. vinifera* cv Chardonnay), as well as a wild *V. vinifera* male (DVIT3351.27) and female genotype (O34-16). In each of these studies, a single representative sex genotype was used to evaluate transcript levels for SDR genes. In our manuscript, we have modified the text and summarized the expression results of *VviPLATZ1* from the flowers of the female, hermaphrodite and male genotypes. “The recessive female flower identity mutation is predicted to reduce the expression and/or function of a candidate gene required for stamen development. Of the seven genes located in the female flower identity region, transcripts for *VviPLATZ1* appeared to correlate with the formation of female flowers^{27,28,30}. Despite the results from these expression studies, a role for the loss of *VviPLATZ1* function in female flower identity was not recognized. Therefore, to support the hypothesis that loss of *VviPLATZ1* results in the formation of female flowers, transcript levels for this gene was examined in 04C023V0006 (*H/H*) and compared to 04C023V0003 (*f/f*) at eight, six and four week prior to anthesis (WPA), as previously defined³⁸” (lines 245-252). Therefore, the expression data to support a role for the loss of *VviPLATZ1* function in female flower development was derived from three hermaphrodites, three females and two males. It should also be pointed out that our expression analyses (RT-qPCR and in situ hybridization) were performed in light of stamen development, rather than flower development as performed by Ramos et al., 2014 and Massonnet et al., 2020.

4. To further address the comments of reviewers #1, we included supplementary figure 1, which compared *VviPLATZ1* transcript levels between 04C023V0003 (*f/f*) and 03C003V0060 (*M/f*). As stated in the manuscript, “In addition, *VviPLATZ1* mRNA levels were readily detected in male flowers of 03C003V0060 (*M/f*) in comparison with 04C023V0003 during stamen development, eight and six WPA (Supplementary Fig. 1)” (lines 258-261). It should be pointed out that 03C003V0060 (*M/f*) was derived from a cross between the female 00C001V0008 microvine x Richter 110 (*V. berlandieri* cv. Boutin B [female] x *V. rupestris* cv. du Lot [male]). Given the pedigree of Richter 110, the male *VviPLATZ1* allele is derived from *V. rupestris*, a distant relative of *Vitis vinifera*. In addition, we included supplementary figure 4, which showed that *VivPLATZ1* is expressed in developing stamens (filaments and anthers), as well as microspores, in 03C003V0060. The following statement in the manuscript refers to supplementary figure 4, “A similar pattern of expression was detected in flowers derived from the male microvine 03C003V0060 (Supplementary Fig. 4).” (lines 325-326). As the *VviPLATZ1* stamen expression pattern in developing flowers of 04C023V0006 (*H/H*) and 03C003V0060 (*M/f*) is conserved between two distantly related *Vitis* species, we believe this is sufficient evidence to suggest that *VviPLATZ1*

is a candidate gene that regulates stamen development during female flower formation.

5. The low expression of VviPLATZ1 in stamens of female flowers and the phenotype of the vviplat1 mutants constitute new findings. As mentioned above, a comparison of gene expression in flower tissues should include several individuals carrying flowers with erect stamens, including male and hermaphrodite individuals.

Response:

We are pleased reviewers #1 recognizes the new findings with our work. We have included supplementary figure 1 and 4, in which the expression levels and patterns of VviPLATZ1 was determined in male flowers in comparison with female flowers (see above). Note: the male VviPLATZ1 allele was derived from *V. rupestris*, a wild North American *Vitis* species. Given the conserved expression pattern of VviPLATZ1 in 04C023V0006 (H/H) and 03C003V0060 (M/f) developing stamens (filaments, anthers and microspores) supports the hypothesis that loss of VviPLATZ1 is a new candidate gene involved in female flower formation in *Vitis*.

6. It has been hypothesized that reflexed filaments in female flowers are due to the smaller size of the epidermal cells on the abaxial (dorsal) side compared to the adaxial side (Dorsey, 1914). Thus, it would be relevant to indicate if differences in filament tissue were observed between hermaphrodite and female flowers at the stage used for in situ hybridization of VviPLATZ1 mRNA, i.e. 6 weeks before anthesis.

Response:

While the histological studies of Dorsey 1914 are interesting, understanding how VviPLATZ1 regulates cell division and expansion during stamen development is beyond the scope of this study.

7. Finally, claims on the impact of VviPLAZ1 on pollen viability should be supported by experiments that test pollen germination of the vviplat1 mutants compared to the control 04C023V0006 (H/H),

Response:

We do not make the claim that VviPLATZ1 impacts pollen viability; however, we speculate that this transcription factor may play a role in pollen development. To address reviewers #1 comments, we have modified the text. "However, VviPLATZ1 may also play a role in pollen function given its location within the female identity region of the SDR (Fig. 1a, b), expression in microspores (Fig. 2c, f), and possible role in regulating the transcription of tRNAs and 5s RNA, as protein synthesis appears to be a primary driver of pollen germination⁵²." (lines 419-422)

Detailed comments

Title

- Page 1, line 1: “Grapevine flower sex determination is controlled by the VviPLATZ1 transcription factor”. The results presented in the study, particularly the phenotype of the vviplat1 mutants, suggest a role in the straightness/erectness of the grapevine flower stamens. However, results do not demonstrate the role of VviPLATZ1 in male fertility as claimed repeatedly in the manuscript. Therefore, the title is catchy, but misleading, and needs to be modified.

Response:

The title has been modified as suggested by reviewer, “**A major determinant of grapevine flower sex identity is controlled by the VviPLATZ1 transcription factor**” (lines 1-2). We have modified the text to reflect that fact that VviPLATZ1 is a major determinant of flower sex identity, as the reflex stamen is the morphological characteristic of female flowers and used by breeders and biologists to phenotype female flower sex.

Introduction

- Page 2, line 52: “While pollen derived from female flowers is viable, fertility is reduced due to an impairment in germination¹⁷”. In their study, Caporali et al. (2002) mentioned that female flowers produce sterile pollen. “Sterile microspores and pollen grains in female flowers display an abnormal round shape, lacking colpi and possessing uniformly thickened cell walls that impede germination.” We suggest revising the sentence to explain that male sterility in female flowers is due to the absence of pollen germination, not a reduction.

Response:

To address this comment by reviewers #1 the sentence was changed to:

“While pollen derived from female flowers is viable, fertility is absent due to an impairment in germination¹⁸ (lines 75-77).”

However, the use of “reduced” is consistent with the view of reviewers #3 who state that “Sometimes pollen from wild female plants is viable and produces offspring.”

- Page 2, line 57: “SEX LOCUS (SL)”. We would suggest naming this region “sex-determining region” in accordance with the recently published studies (Massonnet et al., 2020; Badouin et al., 2020; Zou et al., 2021).

Response:

We have modified the text in the document by changing “SEX LOCUS (SL)” to SEX DETERMINING REGION (SDR) as specified by Zhou et al., 2019, Massonnet et al., 2020, and Zou et al., 2021.

- Page 3, line 78: “As VviYABBY3 is located outside and adjacent to the SL, it is unclear if this gene is the primary M-factor that specifies male flower identity²⁶”. This is another statement not supported by the literature. VviYABBY3 is located inside the sex-determining region according to two recent studies (Massonnet et al., 2020; Zou et al., 2021).

Response:

We have modified the text in the introduction to present the candidate male and female sex factors including VviYABBY3 proposed by Massonnet et al., 2020 and Zou et al., 2021.

“Through sequence analysis of *M*-, *H*- and *f*-haplotypes derived from domesticated and wild *Vitis* species, the structure of the *SDR* was determined and candidate sex determination genes were identified^{27,28,30,31}. Genetic association analysis of *M*- and *f*-specific DNA polymorphisms and phylogenetic studies of sex-linked genes indicates that male and female flower identity is specified by genetic determinants located in the 5' and 3' region of the *SDR*, respectively^{28,31}. Results indicate that the male identity region consists of two genes, *VviYABBY3* and *VviSKU5*, while 11 genes from *TREHALOSE-6-PHOSPHATE PHOSPHATASE (TPP)* to *WRKY* reside in the female identity region³¹. Sequence analysis indicates that the *H*-haplotype is structurally similar to the *f*-haplotype from *VviYABBY3* to the aldolase gene but structurally related to the *M*-haplotype from *TPP* to the *WRKY* transcription factor^{28,31}. Initial studies suggested that *VviAPT3* was the candidate *M*-factor that specifies male flower identity through the inactivation of cytokinin^{25,27,30,32}, which is required for pistil development³³. In support for these studies, transcript levels for *VviAPT3* is significantly increased in male flowers compared to hermaphrodite and female flowers^{27,28,30} and the mRNA for this gene localizes to cells in the male flower meristem that gives rise to the pistil³². Consistent with the ectopic expression of *VviAPT3* in the male flower meristem, applications of cytokinin restores pistil development³⁴. However, recent studies indicate that *VviAPT3* is not located in the sex-linked region that specifies male flower identity³¹. Therefore, *VviYABBY3* has emerged as the primary candidate *M*-factor^{28,31} as this is one of two genes located in the male flower identity region and the expression of this gene correlates with the formation of male flowers. Further, *YABBY* transcription factors are critical for the development of lateral organs, including carpels³⁵. Genetic studies indicate that a candidate DNA polymorphism located in *Vitis INAPERTURATE POLLEN1 (VviINP1)* is the underlying female sterility determinant, as this gene is predicted to play a role pollen development^{27,28,30,36}. While candidate flower sex genes have been identified from the above studies, functional analysis demonstrating a role for these genes in flower sex identity have yet to be determined.” (lines 90-121)

Results

- Page 3, line 94: The genome of PN40024 is not appropriate as reference for detecting SNPs associated with flower sex trait, because the sex-determining region on chr2 is a mixture of H and f haplotypes. We would suggest using a H or f haplotype of the region as reference for the SNP analysis.

Response:

We utilized the Cabernet Sauvignon H- and f-haplotype sequences to verify the 21 *SDR* SNPs identified by comparing the PN40024 and Pinot Noir *SDRs*, see supplementary table 2. In this table, the location of the SNPs is shown for PN40024, as well as the *f*-haplotype from Cabernet Sauvignon. We have also modified the methods section to reflect this information:

“**DNA marker development and genetic mapping.** Sequences for the SSR markers, UDV027, VVIB23, VMC3B10 and VMC6F11, flanking the *SDR* previously identified^{21-23,54}, were used to identify this locus in the 8X PN40024⁵⁵ and Pinot Noir⁵⁶ genomes. Alignment of the 8X PN40024 and Pinot Noir *SDR*-DNA sequences identified 21 single nucleotide polymorphisms (SNPs) spanning this locus (Supplementary Table 2). These SNPs were validated by comparing *H*- and *f*-haplotype sequences in Cabernet Sauvignon (Supplementary Table 2) previously described³⁰.” (lines 467-473)

In light of Nature's policy about data availability, we are concerned about the long-term data availability of grape genomic databases, including grapegenomics.com. NCBI is a long-term database system for genomics data. While grapegenomics.com is extremely useful website for the grape community, the authors wonder about the long-term viability of the website. Previous grapevine genomic databases including VitisExpDB and <https://genomics.research.iasma.it> are defunct and <https://genomes.cribi.unipd.it> is no longer accessible as the connection is not private. Therefore, we request to maintain the positions of SNPs in PN40024.

- Page 3, line 107: "Together, these results are consistent with genetic studies showing that the 5' boundary of the SL is located in TPP25,26." In absence of male individuals in the study, the identified region cannot correspond to the entire sex-determining region. This is a critical flaw of the study.

Response:

We have removed this sentence from the manuscript, as our study is focused on the genetic determinant of female flower identity, not male flower identity.

- Page 3, line 115: "Transcript levels for *VviPLATZ1* was examined in 04C023V0006 (H/H) and compared to 04C023V0003 (f/f) at -8, -6 and -4 stages of flower development previously defined³⁵." To help the reader, we would suggest explaining that -6 stage corresponds to flowers at 6 weeks post-anthesis, for instance. Pictures of the inflorescences at the collection time would also help.

Response:

Reviewers #1 make a good point, so we have modified the text to increase the clarity of the flower developmental stages.

"Therefore, to support the hypothesis that loss of *VviPLATZ1* results in the formation of female flowers, transcript levels for this gene was examined in 04C023V0006 (H/H) and compared to 04C023V0003 (f/f) at eight, six and four week prior to anthesis (WPA), as previously defined³⁸. Results showed that transcripts for *VviPLATZ1* were readily detectable in 04C023V0006 flowers at eight and six WPA (Fig. 1c), which correspond to a developmental time in which stamens differentiate and filaments elongate, respectively³⁸. At four WPA, when flower development is nearly completed, the mRNA levels for this gene was significantly decreased. In contrast to 04C023V0006, transcript levels for *VviPLATZ1* were significantly reduced in 04C023V0003 during stamen development at eight and six WPA (Fig. 1c). In addition, *VviPLATZ1* mRNA levels were readily detected in male flowers of 03C003V0060 (M/f) in comparison with 04C023V0003 during stamen development, eight and six WPA (Supplementary Fig. 1)." (lines 249-261)

We have included images of inflorescences bearing flowers at eight, six and four WPA (Supplementary Fig. 8).

- Page 4, line 127-134: We suggest comparing the mutation events identified in this study with the ones described in studies recently published (Massonnet et al., 2020; Badouin et al., 2020; Zou et al., 2021).

Response:

We have addressed the suggestion by reviewers #1 by comparing female SNPs in 04C023V0003 (*f/f*) to sequenced domesticated and wild *Vitis* female *VviPLATZ1* alleles and the results are shown in supplementary fig. 2 and 3. In addition, we have modified the text in the results section to summarize the results of the alignments.

“DNA sequence comparisons between 04C023V0006 (*H/H*) and 04C023V0003 (*f/f*) identified female specific DNA polymorphisms in *VviPLATZ1*. First, a T to A point mutation and adenine insertion were identified in the first codon, resulting in a frame-shift mutation that altered the position of the start codon (Fig. 1d). In addition, two SNPs were identified at -5 and -13 in the 5' UTR. Through the alignment of hermaphrodite, male and female *VviPLATZ1* alleles from sequenced haplotypes of domesticated and wild *Vitis* species^{28,30}, results showed that female DNA polymorphisms at -13, -5 and +2 were conserved (Supplementary Fig. 2). However, the adenine insertion at +3 in the 04C023V0003 *VviPLATZ1* allele was found only in a subset of female *VviPLATZ1* alleles. Taken together, the conserved female specific DNA polymorphisms at -13, -5 and +2 may act to reduce transcription and/or alter mRNA decay of the female *VviPLATZ1* allele. In addition to these DNA polymorphisms, seven nonsynonymous SNPs were identified in the 04C023V0003 *VviPLATZ1* allele, which may reduce protein function, including an E to K substitution in the PLATZ domain (Fig. 1d). To investigate whether these seven nonsynonymous SNPs were conserved, the 04C023V0003 *VviPLATZ1* amino acid sequence was aligned with hermaphrodite, female and male *VviPLATZ1* proteins derived from sequenced domesticated and wild *Vitis* species^{28,30}. Results showed that all seven nonsynonymous SNPs were conserved in the female *VviPLATZ1* alleles (Supplementary Fig. 3).” (lines 265-282).

- Page 4, line 129: “First, a T to A point mutation and adenine insertion were identified in the first codon, resulting in a frame-shift mutation that altered the position of the start codon (Fig. 1d).” The described mutations would shorten the *f* protein of 2 amino acids. What would be the impact of this shortening on the protein function?

Response:

The text has been modified to increase clarity regarding the significance of the conserved SNPs in the 5' UTR and start codon.

“Through the alignment of hermaphrodite, male and female *VviPLATZ1* alleles from sequenced haplotypes of domesticated and wild *Vitis* species^{28,30}, results showed that female DNA polymorphisms at -13, -5 and +2 were conserved (Supplementary Fig. 2). However, the adenine insertion at +3 in the 04C023V0003 *VviPLATZ1* allele was found only in a subset of female *VviPLATZ1* alleles. Taken together, the conserved female specific DNA polymorphisms at -13, -5 and +2 may act to reduce transcription and/or alter mRNA decay of the female *VviPLATZ1* allele.” (lines 269-275)

- Page 4, line 135-136: “As the phenotype is recessive, the female allele is referred to as *Vviplat1*.” Lower case is commonly used for naming unfunctional allele (as used by the authors for naming knockout mutants). No deleterious mutations were identified in the *f* allele of *VviPLATZ1* and the *f* version of the protein *VviPLATZ1* was not showed to be unfunctional. We would suggest naming *VviPLATZ1* alleles according to their sex-determining haplotypes. i.e. *f* allele of *VviPLATZ1* and *H* allele of *VviPLATZ1*.

Response:

We politely disagree with reviewers #1. It is well established in the grapevine literature that the female allele is recessive. Inheritance studies by Hedrick & Anthony and Valleau, 1916 support this finding. Further, Oberle, 1938 proposed that the female sex determinant is inherited as a recessive trait termed *so*. In addition, our studies also show that the reflex stamen phenotype is a recessively inherited trait (Chaïb et al., 2010). Further, we have shown by functional analysis, that plants that are homozygous for gene edited alleles of *VviPLATZ1* display the reflex stamen phenotype. Therefore, we refer to the female allele as recessive, *Vviplatz1*.

- Page 4, line 138-141: “To gain insight into the proposed function of *VviPLATZ1* in stamen development, the expression pattern of this gene was analyzed in 04C023V0006 (H/H) and 04C023V0003 (f/f) flowers at stage -6 of flower development.” Which criteria were used to select the -6 developmental stage for in situ hybridization of *VviPLATZ1* mRNA? Were differences between female and hermaphrodite filaments observed?

Response:

The text has been modified to explain why in situs were performed at 6 weeks prior to anthesis.

“To gain insight into the proposed function of *VviPLATZ1* in stamen development, the expression pattern of this gene was analyzed in 04C023V0006 (H/H) and 04C023V0003 (f/f) flowers during filament elongation at six WPA. At this time point, *VviPLATZ1* transcripts are significantly higher in 04C023V0006 flowers compared to 04C023V0003 flowers (Fig. 1c).” (lines 289-323)

- Page 4, line 142-143: “A similar pattern of expression was detected in male flowers (data not shown).” Not showing these data is not acceptable. Data from male individuals need to be shown to demonstrate the difference of expression between erect and reflexed filaments. In addition, we suggest including several individuals with different genetic background per sex type to show the low expression of *VviPLATZ1* in stamen tissues of female flowers across the *Vitis* genus.

Response:

To address the comment by reviewers #1, the results in which *VviPLATZ1* transcript levels were compared between 04C023V0003 (f/f) and 03C003V0060 (M/f) are displayed in supplementary fig. 1. The text in the results section was modified to reflect this addition.

“In addition, *VviPLATZ1* mRNA levels were readily detected in male flowers of 03C003V0060 (M/f) in comparison with 04C023V0003 during stamen development, eight and six WPA (Supplementary Fig. 1).” (lines 258-261).

We also included supplementary fig. 4, which displays the *VviPLATZ1* in situ hybridization pattern in 03C003V0060 (M/f) in comparison with 04C023V0003 (f/f). The text of the results section was modified to include this additional supplementary figure.

“In hermaphrodite flowers, *VviPLATZ1* was primarily expressed in elongating filaments, as well as developing anthers and microspores (Fig. 2a-c). A similar pattern of expression was detected in flowers derived from the male microvine 03C003V0060 (Supplementary Fig. 4).

In contrast, *VviPLATZ1* expression was low in female filaments and anthers, as well as microspores (Fig. 2d-f)." (lines 323-327).

The reviewer suggests to perform *VviPLATZ1* in situ hybridization in several individuals with different genetic backgrounds per sex type to show the low expression of *VviPLATZ1* in stamen tissues. First, in our work, we have compared the expression levels and patterns of *VviPLATZ1* in the following microvines, 04C023V0006 (*H/H*), 04C023V0003 (*f/f*) and 03C003V0060 (*M/f*). The microvines 04C023V0006 (*H/H*), 04C023V0003 (*f/f*) were derived from a cross between Grenache x L1 microvine (Pinot Meunier). Therefore, the hermaphrodite and female *VviPLATZ1* alleles in 04C023V0006 (*H/H*), 04C023V0003 (*f/f*) were derived from Grenache and Pinot Meunier. Second, the male microvine 03C003V0060 was derived from a cross between 00C001V0008 x Richter 110. The 00C001V0008 microvine was derived from the selfing (S1) of L1 Pinot Meunier. Based on the parentage of Richter 110 (*V. berlandieri* cv. Boutin B [female] x *V. rupestris* cv. du Lot [male]), the *M*-factor is derived from *V. rupestris* cv. du Lot.

Given that the evolutionary distance between *V. vinifera* and *V. rupestris* together with the conserved *VviPLATZ1* expression patterns in 04C023V0006 (*H/H*) and 03C003V0060 (*M/f*), we believe these results support the conclusion that *VviPLATZ1* is expressed in filaments, anthers and microspores during stamen development. Together, the expression studies and location of *VviPLATZ1* in the female identity region of the *SDR* indicates that loss of *VviPLATZ1* is a new candidate factor that controls female flower identity.

• Page 4, line 168-171: "The average length of stamens in hermaphrodite individuals was 3.0 mm. In contrast, female flowers initiated reflexed stamens 98% of the time (Fig. 4b, g). The remaining 2% of female stamens displayed an upright growth pattern but were significantly shorter with an average length of 1.2 mm." Data are not provided, nor the statistical significance. How many flowers per plant were measured?

Response:

In response the reviewers #1, we have included supplementary fig. 5, which statistically shows that the stunted stamens produced in 04C023V0003 (*f/f*) and the representative gene edited mutant, *Vviplat1-8*, are significant shorter than stamens produced in 04C023V0006 (*H/H*).

• Page 4, line 173: "The morphology of *Vviplat1-1/2*, *-4*, and *-6* were also similar to female flowers (data not shown)." We are not sure why authors don't provide this critical information that can be easily added, if not to the manuscript, at least to supplementary files.

Response:

In response to the comment from reviewers #1, we have included supplementary fig. 6, which displays the flower phenotypes of *Vviplat1/2*, *-4* and *-6*. It should be pointed out that data shown in fig 4g, summarizes stamen morphology in all *Vviplat1* gene-edited plants and results show that this phenotype is highly similar to stamen structure in 04C023V0003 (*f/f*).

Methods

- Page 6, line 240: Methods of SNP genotyping needs to be provided.

Response:

The method of SNP genotyping is described in the “**DNA marker development and genetic mapping section**”.

“Using the Sequenom MassARRAY iPLEX platform (Sequenom, San Diego, CA, USA) service at the Australian Genome Research Facility (AGRF; <http://www.agrf.org.au/services/genotyping>), the genotype for each of the 21 SNPs was determined in F₁ 00C001V0008 x Ugni Blanc individuals, self-pollinated microvines and *Vitis vinifera* cultivars shown in **Supplementary Table 1**. Sequenom primer sequences used for genotyping is shown in **Supplementary Table 3**.” (lines 475-480)

- Page 6, line 251: Authors should indicate if biological replicates were used for evaluating VviPLATZ1 expression.

Response:

We have modified the legend for fig. 1 to include the number of biological replicas.

“Significant differences determined by Student’s *t*-test indicated by asterisks ** $p = 0.014$, * $p = 0.021$ ($n = 3$).” **lines 650-651.**

Reviewer #2 (Remarks to the Author):

This ms by locco-Corena et al. is a contribution to identifying the sex-determining genes in grapevine. Sex determination is much less known in plants than in animals, but these 5-6 years great advances have been made in finding plant sex-determining genes. Two papers on a genomic characterization of the grapevine sex locus came out last year and several good candidate sex-determining genes were pointed out (Badouin et al. 2020, Massonnet et al. 2020). No functional validation however was carried out. This ms reports a Crisp-cas9 validation of the VviPLTAZ1 gene using microvines. They showed that this gene underlies reflex stamens in females, an important aspect of floral morphology in grapevine that was left unexplained in previous work (Badouin et al. 2020, Massonnet et al. 2020).

I found the use of microvines and Crisp-cas9 very elegant. This is an important contribution to understanding grapevine sex determination. It should be noted that functional validation of sex-determining genes is rarely done in plants since not all plants can be easily transformed, which makes this contribution also important for the whole field of plant sex determination. I am thus very positive about this work. I however have several comments:

1) The introduction of the ms includes a discussion of the papers I mentioned above (Badouin et al. 2020; Massonnet et al. 2020). I did not think that these papers were very well presented. Both papers agreed on VviINP1 being a male-determining gene. They however differed for the female-determining one. Badouin et al. suggested VviAPT3 was the female-determining gene, Massonnet et al. suggested VviYABBY3 was the one. I think this part of the introduction should be re-written.

Response:

The paragraph referred by reviewer # 2 has been modified to increase clarity about the organization of the SDR and candidate genes implicated in sex determination.

“Through sequence analysis of *M*-, *H*- and *f*-haplotypes derived from domesticated and wild *Vitis* species, the structure of the SDR was determined and candidate sex determination genes were identified^{27,28,30,31}. Genetic association analysis of *M*- and *f*-specific DNA polymorphisms and phylogenetic studies of sex-linked genes indicates that male and female flower identity is specified by genetic determinants located in the 5' and 3' region of the SDR, respectively^{28,31}. Results indicate that the male identity region consists of two genes, *VviYABBY3* and *VviSKU5*, while 11 genes from *TREHALOSE-6-PHOSPHATE PHOSPHATASE (TPP)* to *WRKY* reside in the female identity region³¹. Sequence analysis indicates that the *H*-haplotype is structurally similar to the *f*-haplotype from *VviYABBY3* to the aldolase gene but structurally related to the *M*-haplotype from *TPP* to the *WRKY* transcription factor^{28,31}. Initial studies suggested that *VviAPT3* was the candidate *M*-factor that specifies male flower identity through the inactivation of cytokinin^{25,27,30,32}, which is required for pistil development³³. In support for these studies, transcript levels for *VviAPT3* is significantly increased in male flowers compared to hermaphrodite and female flowers^{27,28,30} and the mRNA for this gene localizes to cells in the male flower meristem that gives rise to the pistil³². Consistent with the ectopic expression of *VviAPT3* in the male flower meristem, applications of cytokinin restores pistil development³⁴. However, recent studies indicate that *VviAPT3* is not located in the sex-linked region that specifies male flower identity³¹. Therefore, *VviYABBY3* has emerged as the primary candidate *M*-factor^{28,31} as this is one of two genes located in the male flower identity region and the expression of this gene correlates with the formation of male flowers. Further, YABBY transcription factors are critical for the development of lateral organs, including carpels³⁵. Genetic studies indicate that a candidate DNA polymorphism located in *Vitis INAPERTURATE POLLEN1 (VviINP1)* is the underlying female sterility determinant, as this gene is predicted to play a role pollen development^{27,28,30,36}. While candidate flower sex genes have been identified from the above studies, functional analysis demonstrating a role for these genes in flower sex identity have yet to be determined.” (lines 90-121)

2) Previous work disagreed about the boundary of the sex locus. Badouin et al. considered *VviYABBY3* outside of the sex locus, while Massonet et al. considered it inside. Revisiting this question is thus helpful. Iocco-Corena et al. found that both *VviAPT3* and *VviYABBY3* are outside of the sex locus. However, I think that their results should be interpreted more cautiously. First, they have used the 12X version of PN40024, in which the sex locus is not well assembled (*VviAPT3* is missing). This might have affected how they located their markers in the sex locus. This should be discussed. Second, they found a recombination event at the *VviMT72* marker, but recombination between *f* and *H* haplotypes are known to have happened and some *H* haplotypes have in fact the *f* version of *VviAPT3*. Their results might hold only for cultivated grapevines and not wild ones. Unless, they can show evidence that *VviAPT3* is not genetically linked to the sex locus in wild grapevine, they should make it clear that their interpretation is restricted to cultivated grapevines (and might even not be valid for all cultivated grapevines). This is important as in previous work (in particular in Badouin et al.) the boundaries of the sex locus have been studied in wild grapevines. It

might be confusing for the readers if this point is not clarified and discussed. Also, it means that they cannot really rule out VviAPT3 as being a sex-determining gene (see also my comments 4 and 5).

Response:

As recommended by reviewer #1, we have used the Cabernet Sauvignon *f* and *H* haplotype sequences as reference for the SNP analysis. As a result, we have updated fig. 1a, b and supplementary table 2, as well as the method section (see below).

“DNA marker development and genetic mapping. Sequences for the SSR markers, UDV027, VVIB23, VMC3B10 and VMC6F11, flanking the *SDR* previously identified^{21-23,54}, were used to identify this locus in the 8X PN40024⁵⁵ and Pinot Noir⁵⁶ genomes. Alignment of the 8X PN40024 and Pinot Noir *SDR*-DNA sequences identified 21 single nucleotide polymorphisms (SNPs) spanning this locus (Supplementary Table 2). These SNPs were validated by comparing *H*- and *f*-haplotype sequences in Cabernet Sauvignon (Supplementary Table 2) previously described²⁸.” Lines 467-473.

The focus of the manuscript is not on VviAPT3 but VviPLATZ1. We have genetically shown via the recombination event at VviMT72 in 06C008V0018 excludes VviAPT3, as well as FMO4, VViFSEX and WRKY21.

3) The mutations that they found in VviPLTAZ1 (*f* version, see Figure 2) are very interesting and might explain why its expression is reduced in females. Looking at male individual (of *V. sylvestris*) would have been very interesting to verify that these mutations are absent in the *M* haplotype.

Response:

We have performed an amino acid alignment and compared the 04C023V0003 (*f/f*) VviPLATZ1 allele with hermaphrodite, female and male alleles from domesticated and wild *Vitis* species (see supplementary fig. 3).

“DNA sequence comparisons between 04C023V0006 (*H/H*) and 04C023V0003 (*f/f*) identified female specific DNA polymorphisms in *VviPLATZ1*. First, a T to A point mutation and adenine insertion were identified in the first codon, resulting in a frame-shift mutation that altered the position of the start codon (Fig. 1d). In addition, two SNPs were identified at -5 and -13 in the 5' UTR. Through the alignment of hermaphrodite, male and female *VviPLATZ1* alleles from sequenced haplotypes of domesticated and wild *Vitis* species^{28,30}, results showed that female DNA polymorphisms at -13, -5 and +2 were conserved (Supplementary Fig. 2). However, the adenine insertion at +3 in the 04C023V0003 *VviPLATZ1* allele was found only in a subset of female *VviPLATZ1* alleles. Taken together, the conserved female specific DNA polymorphisms at -13, -5 and +2 may act to reduce transcription and/or alter mRNA decay of the female *VviPLATZ1* allele. In addition to these DNA polymorphisms, seven nonsynonymous SNPs were identified in the 04C023V0003 *VviPLATZ1* allele, which may reduce protein function, including an E to K substitution in the PLATZ domain (Fig. 1d). To investigate whether these seven nonsynonymous SNPs were conserved, the 04C023V0003 *VviPLATZ1* amino acid sequence was aligned with hermaphrodite, female and male *VviPLATZ1* proteins derived from sequenced domesticated and wild *Vitis* species^{28,30}. Results showed that all seven

nonsynonymous SNPs were conserved in the female *VviPLATZ1* alleles (Supplementary Fig. 3)." (lines 265-282)

4) Their expression study of *VviPLATZ1* is interesting and does provide useful data compared to previous work. However, I am not convinced by the interpretation about the expression of this gene in the ovule. They say: « Interestingly, *VviPLATZ1* was expressed in the ovule of hermaphrodite and female flowers (Fig. 2a, d, respectively). The molecular studies above together with the reduced expression of *VviPLATZ1* in developing stamens of female flowers indicates that this gene is involved in flower sex determination. » and in the abstract they say : « Taken together, the results demonstrate that loss of *VviPLATZ1* function is THE major determinant for flower sex determination in *Vitis*. » I think they have shown that *VviPLATZ1* is an important determinant of flower sex determination, I don't think that they have shown it is THE major determinant. The role of *VviPLATZ1* mutation in producing reflex stamens in female flower is clear, the role in preventing ovule production is not clear at all. The above sentences seems to suggest that there is only one sex-determining gene in grapevine with both male and female functions. Classical genetic experiments suggested that sex is determined by two genes in grapevine. I don't think the authors have gathered evidence that this is not true and *VviPLATZ1* is the only determinant (see also next comment). This should be clearly explained in the ms.

Response:

To address the comment and concern raised by reviewer #2, we have modified the title to "**A major determinant of grapevine flower sex identity is controlled by the *VviPLATZ1* transcription factor**" (lines 1-2). To increase the clarity of the manuscript we have modified the text to reflect that *VviPLATZ1* is a major determinant of female flower identity/determination. In the manuscript, we never hypothesized that *VviPLATZ1* is involved in male flower identity.

5) Another important point is about the manipulation of *VviPLATZ1* using Crispr-cas9. The results are very neat, they do show that *VviPLATZ1* is underlying the reflex stamens phenotype. However, differences in flower morpho-anatomy between male and female are more than just reflex stamens. Pollen is infertile in female flowers. This has not been looked at in the Crispr-cas9 modified plants. There is no evidence that *VviPLATZ1* is underlying this floral trait as well.

Response:

The reviewer raises an important point about the possible role of *VviPLATZ1* in pollen development. Determining a role for *VviPLATZ1* in pollen development is beyond the scope of this work. However, we speculate that it is possible that *VviPLATZ1* play a role in pollen development.

"However, *VviPLATZ1* may also play a role in pollen function given its location within the female identity region of the SDR (Fig. 1a, b), expression in microspores (Fig. 2c, f), and possible role in regulating the transcription of tRNAs and 5s RNA, as protein synthesis appears to be a primary driver of pollen germination⁵²." Lines 419-422.

Minor comments :

I think a few words are needed to explain what are the microvines, first time they are

mentioned in the text, for the readers who are not grapevine experts.

Response:

To address the concern of reviewer #2, we have modified the last paragraph in the introduction to explain the novel usage of the microvine in validating the function of *VviPLATZ1* in female flower identity.

“The ability to efficiently validate the function of candidate flower sex identity genes via reverse genetics is constrained by the perennial nature of grapevine, which displays a long juvenile phase and annual reproductive cycle³⁷. Therefore, the role of *VviPLATZ1* in female flower identity was functionally validated using CRISPR/Cas9 system in a hermaphrodite microvine, a genotype with a reduced generation cycle that displays a continuous flowering phenotype that is highly amendable to rapid reverse genetics assessment³⁷.” (lines 128-134).

Reviewer #3 (Remarks to the Author):

Manuscript background information

All grapevine species are dioecious, with the only exception being the cultivated variety, *V. v. vinifera*. Hermaphroditism in cultivated *V. v. vinifera* is a trait that was gain during domestication. In this paper the authors, Iocco-Corena and co-workers, tried to go deeper in a century old question: what makes grapevines dioecious? The authors analysed the female sex locus region and identified the *VviPLATZ1* transcription factor as a candidate flower sex factor. The authors used the CRISPR/Cas9 system to edit *VviPLATZ1* (loss of function) in a hermaphrodite genotype and showed that individual lines homozygous for the gene edited alleles gave rise to female flowers with reflex stamens. An evidence of *VviPLATZ1* is involved in flower sex determination in *Vitis*.

Comments for transmission to the authors

In general, this is an interesting research paper. However, it requires a minor revision of the English language, since in few instances the manuscript lacks fluidity possible due to several authors contributing. The discussion is very well based on results and the methodology seems robust. There are, however, some small questions and remarks.

1 – I do not think references in the abstract is a common occurrence. I would advice removing it from the abstract.

Response:

The references were removed.

2 – The names of the genes are in italics. Please review the article.

Response:

Standard nomenclature was used in the manuscript in which italics were used to represent the gene, *VviPLATZ1*; whereas, references to the protein are no italicized. We have checked

over the manuscript and made appropriate changes in light of this comment of reviewer #3.

3 – In Line 83-85 It was not clear why, among several genes in the SEX LOCUS, the authors decided to focus on VviPLATZ. Could you explain the reason, please? You state that through genetic mapping you identified VviPLATZ1 but no reference is supplied.

Response:

To increase clarity, we have modified this sentence. “Through genetic mapping and expression studies, we identified the *VviPLATZ1* (plant AT-rich sequence-and zinc-binding protein1) transcription factor as a new candidate sex determinant in which reduced expression of this gene results in the formation of female flowers.” lines 125-128

4 – Authors made very clear the role of VviPLATZ in the stamen formation in female grapevine. However, in the plants bearing the edited alleles of PLATZ it would be interesting to assess the expression value of other putative important genes in the female portion of the sex locus, namely VviINP1 and even VviAPRT3, VviYABBY and VviFSEX. Although I understand that this might not be in the scope of the manuscript.

Response:

Reviewer #3 makes an interesting point, but we agree with them that this is beyond the scope of the manuscript.

Line 15

“modern domesticated *Vitis vinifera* cultivars form hermaphrodite flowers capable of self-pollination” please bear in mind that some female cultivars of *V. vinifera* exist for wine production making this phrase not entirely true.

Response:

We are aware that some female cultivars exist or existed for wine production, which is why we used “modern domesticated cultivars”. Lines 16 and 69-70.

Line 18

“To identify the recessive allele responsible for the development of reflex stamens” I do not agree with the use of the word “recessive” in the abstract, merely because the authors haven’t yet explained why it should be a recessive allele. Although scientists familiarized with the *Vitis* sex dilemma recognize this fact other researchers may not have this perception.

Also, in the abstract the authors claim that VviPLATZ is the major player for flower sex determination (Line 27). The use of the term “major” implies knowledge of the function of the other genes in the sex region. At the moment, this knowledge is not yet available. Therefore, I recommend some caution when using this term.

Response:

As recommended by reviewer #3, we have removed “recessive” from the abstract. We understand the reviewers’ advice about using the word “major”; however, the two-gene model of sex determination is widely accepted. Genetic studies used to identify the female

sex determinant was based on the phenotyping individuals that produce female flowers, which is characterized by the formation of reflex stamens. We show in our functional analysis that gene edited VviPLATZ1 give rise to reflex stamens. If there is another female determinant involved in the formation of female flowers, that would indicate that the two-gene model is no longer valid. Furthermore, given that all seven alleles of VviPLATZ1 give rise to flowers with reflex stamens that is indistinguishable from the flowers produced by female vines, demonstrates that loss of VviPLATZ1 is a major determinant of female flower identity.

Line 28-29

“Traits associated with flower development and fertilization are key for increasing yield in agriculture and horticulture crops” – either a reference is missing or please replace “are” for “could be”.

Response:

We have referenced this sentence as suggested.

“Traits associated with flower development and fertilization are key for increasing yield in agriculture and horticulture crops².” **Lines 52-53**. The reference is Manrique, S. et al. Genetic insights into the modification of the pre-fertilization mechanisms during plant domestication (2019).

Line 33

“...underlying mechanisms that control flower identity are distinct.” – A reference is missing at the end of this sentence.

Response:

The information in this sentence is referenced in the previous sentence, **lines 54-55**.

Line 46-48

Please bear in mind that several female cultivars of *V. vinifera* exist and are used for wine production. This means that, *V. vinifera* cultivars for wine are not exclusively hermaphrodites.

Response:

We have changed the phrase “Cultivated grapevines” to “modern cultivated grapevines” to reflect the fact that a several female cultivars exist (see above).

Line 80

I think is important the authors to refer the putative importance of VviINP1 and the 8bp deletion potentially involved in pollen infertility in female plants and VviFSEX which, although some groups show a non-specific location of mRNA and non-differentially expressed (Coito et al., 2017) other groups showed to be differentially expressed (Zhou et al 2017, Badouin et al., 2020).

Response:

We referred to importance of *VviINP1* on **lines 117-119 and 413-419**. However, tissue specific expression and functional analyses are required to validate *VviINP1* as a major sex

determinant of pollen fertility. We did not consider *VviFSEX* as a female candidate factor, since this gene is not localized the female identity region identified in our manuscript and in situ analysis performed in Coito et al., 2017 showed that this gene is expressed in stamens and carpels of male, female and hermaphrodite flowers. Therefore, the expression pattern of this gene does not appear to correlate with stamen and/or pollen development.

Line 98

Out of curiosity could you state the germination rate of the crosses?

Response:

Seed germination rates were 24-59% for microvines, similar to rates observed in *V. vinifera* (Chaïb et al., 2010)

Line 103-104

“genotyping self-fertilized hermaphrodite microvine (H/f) progenies (S1 to S5)”, could you provide the segregation rate? According to my calculation it should results in a 3:1 hermaphrodite to female rate (or a 2:1:1 of h/f, h/h and f/f).

Response:

Yes, segregation ratio for hermaphrodite and female flowers in self-fertilized microvines was 3:1, as this phenotype is recessive (Oberle 1938; Antcliff, 1980). We have modified the following sentence to reflect this. “Self-fertilized hermaphrodite microvine (H/f) progenies (S1 to S5) segregating for hermaphrodite and female individuals with a 3:1 ratio were also genotyped with the SDR SNP set.” Lines 152-154

Line 104-105

Out of curiosity, how does the SNP set used links with *VviAPRT3* and *VviFSEX* from Coito et al., 2017 and *VviINP1* of Badouin et al., 2020 and Massonnet et al., 2020?

Response:

The SDR 21-SNP set was developed before the Coito et al., 2017; Badouin et al., 2020 and Massonnet et al., 2020 manuscripts were published. *VviAPRT3* and *VviFSEX* are located downstream of the female identity region and therefore do not appear to play a primary role in specifying female flower identity. Please refer to Fig 1b and lines 143-243 in the results section.

Line 109-111

Before you stated that the SL contains 11 genes. However, you narrow the SL to only seven genes. Are these seven genes responsible for the female identity? Please clarify. How does this compare with the recent works of Badouin et al., 2020 and Massonnet et al., 2020?

Response:

The introduction paragraph describing the SDR has been modified to increase clarity regarding the structure of the SDR and gene content.

“Through sequence analysis of *M*-, *H*- and *f*-haplotypes derived from domesticated and wild *Vitis* species, the structure of the *SDR* was determined and candidate sex determination genes were identified^{27,28,30,31}. Genetic association analysis of *M*- and *f*-specific DNA polymorphisms and phylogenetic studies of sex-linked genes indicates that male and female flower identity is specified by genetic determinants located in the 5’ and 3’ region of the *SDR*, respectively^{28,31}. Results indicate that the male identity region consists of two genes, *VviYABBY3* and *VviSKU5*, while 11 genes from *TREHALOSE-6-PHOSPHATE PHOSPHATASE (TPP)* to *WRKY* reside in the female identity region³¹. Sequence analysis indicates that the *H*-haplotype is structurally similar to the *f*-haplotype from *VviYABBY3* to the aldolase gene but structurally related to the *M*-haplotype from *TPP* to the *WRKY* transcription factor^{28,31}.” Lines 90-99.

In addition, we have modified Fig 1a and 1b which allows the reader to compare our results with recent studies that characterized the female identity region in the *SDR*. The summary sentence of the results paragraph describing the genetic mapping was also modified to increase clarity. “Taken together, fine genetic mapping demonstrated that the female flower identity region in the *SDR* contained seven genes: *VviINP1*, *Exostosin*, 3-ketoacyl-acyl carrier protein synthase III (*KASIII*), *VviPLATZ1*, *FMO1*, *FMO2* and *FMO3* (Fig. 1b).” lines 157-243.

In summary, association studies from Zou et al., 2021 indicates that the female identity region consists of 11 genes, from *TPP* to *WRKY*. In our study, we narrowed the female identity region to 7 genes from *INP1* to *FMO3*.

Line 115

Although referenced, I think the manuscript would benefit greatly with representative images of the development stages used: -8, -6 and -4.

Response:

We have modified the text to increase the clarity of the flower developmental stages used in the expression studies, which was performed in light of stamen development in the different flower genotypes (see lines 249-263). In addition, supplementary figure 8 displays the inflorescence morphologies used to identify flowers at 8, 6 and 4 weeks prior to anthesis (WPA).

Line 124-125

I think a correction should take place in this sentence. Badouin et al., 2020 reports a higher expression of *VviINP1* in female flowers.

Response:

We proposed the following in this section of the results: “The recessive female flower identity mutation is predicted to reduce the expression and/or function of a candidate gene required for stamen development. Of the seven genes located in the female flower identity region, transcripts for *VviPLATZ1* appeared to correlate with the formation of female

flowers^{27,28,30} (lines 245-248). That is, a candidate female flower identity gene is expected to display lower expression levels in female flowers compared to hermaphrodite and male flowers. Results from Badouin et al., 2020 and Massonnet et al., 2020 show that transcripts for *VviINP1* are higher in female flowers at late stages of flower development. Therefore, we concluded that “Transcript levels for *VviINP1*, *Exostosin*, *KASIII*, *FMO1*, *FMO2* and *FMO3*, which are located in the female flower identity region of the SDR, did not correlate with the formation of female flowers^{27,28,30} (lines: 261-263).

Line 127 to 132

The *VviPLATZ* gene that the authors refer to is not the one located on chromosome2 in the database referred to in Fig1 (grapegenomics.com). This gene with the described sequence is located on the unknown chromosome. I would suggest that the authors contextualize these locations and why you chose to work with the unknown chromosome allele

Response:

As stated on lines 127-128 of the submitted article (and revised article, lines 265-266), “DNA sequence comparison between 04C023V0006 (H/H) and 04C023V0003 (f/f) identified female specific DNA polymorphisms in *VviPLATZ1*.” As written in the methods, the hermaphrodite and female alleles were identified by cloning and sequencing *VviPLATZ1* cDNAs from 04C023V0006 (H/H) and 04C023V0003 (f/f). “To clone *VviPLATZ1*, the B26 primer (GACTCGAGTCGACATCGATTTTTTTTTTTTTTTTTT) was used to prime cDNA synthesis in the 04C023V0006 and 04C023V0003 cDNA samples derived from flowers at six WPA. Subsequently, *VviPLATZ1* mRNA was amplified using primers CSPla1_CDS_F1 (CAGTGCCAGTTTTGCAGGC) and B25 (GACTCGAGTCGACATCGA). After subcloning the PCR products into the pDRIVE vector using the Qiagen PCR Cloning Kit (Qiagen, Chadstone, VIC, Australia), Sanger sequencing was performed at the AGRF to determine the sequence of *VviPLATZ1* in 04C023V0006 and 04C023V0003.” (lines 500-507).

In the new version of the manuscript, we aligned the 04C023V0006 (H/H) and 04C023V0003 (f/f) *VviPLATZ1* alleles with female, hermaphrodite and male alleles derived from wild and domesticated *Vitis* species. Therefore, we haven’t chosen to work with an unknown chromosome allele.

Line 144-145

Why do the authors consider this observation interesting? What about male plants? Do they also show *VviPLATZ* mRNA in the ovule? If only hermaphrodites and female plants show this presence, then *VviPLATZ* might play a more complex role in flower formation. From my understanding male plants still display ovules.

Response:

We modified the sentence that reviewer #2 commented on to:

“The expression of *VviPLATZ1* in ovules of hermaphrodite and female flowers (Fig. 2a, d, respectively), as well as male flowers (Supplementary Fig. 4) suggests that this gene may play a role in ovule development” lines 327-330. At this time, we have not assessed whether

seed development is impaired or functional in the gene-edited *Vviplatz1-1/2, 3, 4, 5, 6, 7, 8* plants.

Line 159-161

Despite showing altered stamens all plants bearing the altered PLATZ could still produce viable pollen? Sometimes pollen from wild female plants is viable and produces offspring.

Response:

Yes, we agree with reviewer 3 that female flowers can give rise to functional pollen and viable offspring. We have not assessed pollen viability as addressing this role is beyond the scope of our study. The focus of this study was on identifying genetic determinants of reflex stamen development as this phenotype is a major characteristic of female flower identity.

Line 170-171

Grapevine stamens polymorphism was described previously and so, despite being old, would be nice if could cite the work of Stout, 1921 (<http://www.mavo.biz/Reprints/Stout1921.pdf>) where polymorphism of grapevine stamens is described.

Response:

Thank you for the very interesting paper. We have cited this paper (Lines 369-370).

Line 195

What is the relation of maize floury3 and rice GL6 with *VviPLATZ* family? Are they homologues, orthologues? Please clarify.

Response:

Phylogenetic analysis indicates that *VviPLATZ1* is orthologous with GL6, as well as other PLATZ proteins from maize, rice and Arabidopsis (see supplementary Fig. 7). *VviPLATZ1* is a homologue of FL3. As the other nine PLATZ proteins contained in the GL6 and *VviPLATZ1* clade are uncharacterized, it is difficult to draw any meaningful conclusion from this analysis. However, we have included the following sentence to increase clarity about the evolutionary relationship of *VviPLATZ1* with other PLATZ proteins.

“Lastly, phylogenetic analysis showed that *VviPLATZ1* is a member of a clade that includes GL6, as well as uncharacterized PLATZ proteins from *Zea mays*, *Oryza sativa* and *Arabidopsis thaliana* (Supplementary Fig. 7).” (lines 408-411).

Line 210-211

“As a result, gene editing of *VviINP1* is necessary to validate a role for the 8-bp deletion in pollen fertility.” Just a minor note, but I think the phrase would be better as “As a result, gene editing of *VviINP1*, in grapevine, is necessary to validate a role for the 8-bp deletion in pollen fertility.”

Response:

Thanks for the suggestion. We modified the sentence as suggested above (lines 418-419).

Line 211-213

“However, VviPLATZ1 may also control pollen fertility given its expression in microspores (Fig. 2c, f), close proximity to VviINP1 within the SL (Fig. 1a, b) and role as a transcription factor.” This sentence is not very clear.

But if understand correctly, you self pollinate plants with the edited VviPLATZ. So the pollen was fertile? Was the number of fruits produced in this self-pollination different from a self-pollination of non edited plants?

Perhaps the photos shown in figure 2 are not the best to demonstrate the expression of VviPLATZ in the microspores. The signal is weak in the female plant and the hermaphrodite has some background noise. Perhaps in this case it would be advisable to place the images of the controls even in annex.

Response:

In this sentence, we speculate that VviPLATZ1 may play a role in pollen fertility. We have changed the sentence to clarify its meaning.

“However, VviPLATZ1 may also play a role in pollen function given its location within the female identity region of the SDR (Fig. 1a, b), expression in microspores (Fig. 2c, f), and possible role in regulating the transcription of tRNAs and 5s RNA, as protein synthesis appears to be a primary driver of pollen germination⁵².” Lines 419-422.

As explained in the methods section, “seven first-generation plants (T₀) containing gene edited alleles were identified and self-pollination was performed to evaluate the flower sex phenotype in T₁ plants.” lines 570-571.

VviPLATZ1 is expressed in the filaments and anthers of stamens in 04C023V0006 (*H/H*) and 03C003V0060 (*M/f*), as well as microspores. The “background” staining referred by reviewer #3 is staining in the anther.

Line 226

This section of the methods is a bit confusing. As I understand, F1 plants and S1 to S5 were already established from previous studies (reference 34 and 47) however the way the manuscript is written made me think that you edited VviPLATZ and then self-pollinate the edited grapevine for five generation. I think this should be clarified.

Response:

We have modified the text to increase clarity regarding the plant material used in our study.

“The 04C023V0006 (*H/H*) and 04C023V0003 (*f/f*) microvines, which were derived from a Grenache (*H/f*) x L1 microvine cross (*H/f*), were used for molecular and gene expression analyses. Gene editing was performed in 04C023V0006. Gene expression studies was also performed in 03C003V0060 (*M/f*), which was derived from a 00C001V0008 (selfed L1 Pinot Meunier microvine) x Richter 110 (*V. berlandieri* cv. Boutin B x *V. rupestris* cv. du Lot). Plant material used for fine genetic mapping was derived from F1 00C001V0008 (*f/f*) x Ugni Blanc

(H/f) individuals and successive self-pollinated microvines (H/f) from first (S1) to fifth generation (S5) as previously described³⁷." (lines 432-462)

Line 228-229

"00C001V0008 (f/f) x Ugni Blanc (H/f) individuals and successive self-pollinated microvines²²⁹ (H/f) from first (S1) to fifth generation (S5) as previously described"
The individual 00C001V0008 is a microvine? Is also derived from the Grenache (H/f) x L1 microvine cross? Also how many time between generations elapsed? Were the plants supplied with hormones in order to accelerate the life cycle and produce flowers early?

Response:

Yes, 00C001V0008 is a microvine derived from a selfed L1 Pinot Meunier microvine (lines 438-439). The generation time for microvines is approximately 6 months (Chaïb et al., 2010). No hormones are applied to accelerate the life cycle or induce flowering (Boss and Thomas, 2002).

This article was reviewed by Margarida Rocheta and João Lucas Coito from Instituto Superior de Agronomia, Lisbon, Portugal

Reviewer #4 (Remarks to the Author):

All of the approximately 70 species in the genus *Vitis* are dioecious, and only the cultivated grapevine, *Vitis vinifera* subsp. *vinifera*, has reverted to hermaphroditism, while its wild ancestor, *V. vinifera* subsp. *sylvestris*, retains the dioecious condition. The shift occurred during domestication about 8,000 years ago. Pat Iocco-Corena and colleagues have used genetic mapping and expression studies to identify a candidate transcription factor that regulates stamen architecture during grapevine flower development, which they have christened *VviPLATZ1* (plant AT-rich sequence-and zinc-binding protein1). They used the CRISPR/Cas9 system to edit *VviPLATZ1* in a hermaphrodite genotype (of cultivated wine). Phenotype evaluation showed that individuals with gene-edited alleles of *VviPLATZ1* had flowers with reflexed architecture, that is, flowers that are functionally female because they cannot self-pollinate. *VviPLATZ1* thus is a novel regulator of stamen architecture.

The results are clear, and the study is a great step forward. As the authors point out, a candidate gene suggested to control pistil suppression in male flowers of wild wine, namely *VviAPT3*, and another involved in inaperturate pollen (*VviINP1*), causing male-sterility candidate in wild wine, still need to be tested with the CRISPR/Cas9 system.

There are numerous sentences in which the words 'gene edited' need to be hyphenated: gene-edited *VviPLATZ1* alleles; gene-edited mutations; gene-edited *VviPLATZ1* plants. gene-edited *VviPLATZ1* – plants missing; etc.

Response

We changed "gene edited" to "gene-edited" in the manuscript

Also, search for adn instead of and

Response

The reference containing "adn" has been corrected, thanks.

Reviewers' Comments:

Reviewer #1:

Remarks to the Author:

Main comments

As we mentioned in our past review, the results presented in the manuscript show elegantly that VviPLATZ1 plays a role in stamen architecture. However, the results do not show that VviPLATZ1 is involved in sex determination in grapes. Although reflex stamen is a morphologic trait that is commonly used to evaluate grape flower sex phenotype, female flowers with erect filaments were described in previous studies (Stout, 1921; Cunha et al., 2020; IPGRI, UPOV, OIV, 1997), as well as in this manuscript ("1-5% of the stamens initiated in f/f were stunted and grew to an average length of 1.2"), indicating that reflex stamens are not responsible for male sterility in *Vitis* female flowers. Indeed, self-pollination could still occur through wind or pollinators if the pollen grains were able to germinate. Male fertility could be tested using an in vitro pollen germination assay or by bagging inflorescences before blooming. As the effect of the loss of VviPLATZ1 function on male fertility was not evaluated in this study, the role of VviPLATZ1 in sex determination cannot be stated. Accordingly, we ask again the authors to revise the claim that VviPLATZ1 is a sex-determining gene.

References:

Stout (1921) Stout AB. Types of flowers and intersexes in grapes with reference to fruit development. New York Agricultural Experiment Station Technical Bulletin. 1921;82:1-16.

Cunha, J., Ibáñez, J., Teixeira-Santos, M., Brazão, J., Fevereiro, P., Martínez-Zapater, J. M., & Eiras-Dias, J. E. (2020). Genetic Relationships Among Portuguese Cultivated and Wild *Vitis vinifera* L. Germplasm. *Frontiers in plant science*, 11, 127. <https://doi.org/10.3389/fpls.2020.00127>

IPGRI, UPOV, OIV. 1997. Descriptors for Grapevine (*Vitis* spp.). International Union for the Protection of New Varieties of Plants, Geneva, Switzerland/Office International de la Vigne et du Vin, Paris, France/International Plant Genetic Resources Institute, Rome, Italy. https://www.ecpgr.cgiar.org/fileadmin/biodiversity/More_pubs/393_Descriptors_for_grapevine__Vitis_spp._.pdf

Major reviews

Title

Title is confusing and misleading. It implies that VviPLATZ1 controls a major determinant of grapevine flower sex determination. Authors use the term "determinant" for both genetic/molecular determinant and morphological determinant (line 123 for instance) in the manuscript, which is confusing. We suggest using this term for genetic determinant only. Also, the work doesn't implicate VviPLATZ1 in sex determination, but in stamen morphology and the title should reflect that.

Results

- Line 149-150: "Results showed that the 5' boundary of the SDR that specifies female flower identity was located in TPP, as indicated by the VvMT_54 SNP (Fig. 1b)." Figure 1 a & b depict the f haplotype of Cabernet Sauvignon, but SNP calling was performed using the PN40024 haploid genome as reference. As previously mentioned, PN40024 genome is not appropriate as reference for studying the SDR. The genome of PN40024 is not appropriate as reference for detecting SNPs associated with flower sex trait, because the sex-determining region on chr2 is a mixture of H and f haplotypes (see Zhou et al., 2019 *Nature Plants*). Therefore, we ask the authors to use a H or f haplotype of the SDR region as reference for the SNP analysis. One option is to use Cabernet Sauvignon, which as described in Massonnet et al. (2020) is available on a permanent repository (<https://doi.org/10.5281/zenodo.3827985>). Another option is to use the female *syvestris* C1-2, whose reads, genome assembly, and gene annotation are available in ENA under the bioproject PRJEB37020" (Badouin et al., 2020).

- Lines 258-261: It would be pertinent to present the results of VviPLATZ1 expression in flowers from the male individual 03C003V0060 in Figure 1c because VviPLATZ1 seems to be lower expressed in the male flowers compared to the hermaphrodite ones (Supplementary Fig. 1). If so, it should be discussed.

- Line 262: "female flower identity region of the SDR". Floral organ-identity genes are the genes specifying floral organ identity. Therefore, "female flower identity region" should be named the female-associated region.

Methods

- Lines 469-471: "Alignment of the 8X PN40014 and Pinot Noir SDR-DNA sequences identified 21 single nucleotide polymorphisms (SNPs) spanning this locus". First, PN40024 genome should not be used for studying sex determination. Second, how was the alignment performed and how were SNPs identified?

- Lines 471-473: "These SNPs were validated by comparing H- and f-haplotype sequences in Cabernet Sauvignon (Supplementary Table 2) previously described²⁸" In Supplementary table 2: "1 From <http://www.grapegenomics.com>. PN40024_Chr02_V2.1. 1 From <http://www.grapegenomics.com>. f-haplotype of Cabernet Sauvignon." The number "1" indicates for both PN40024 and Cabernet Sauvignon genomes. How were the H and f haplotype compared? How were SNPs identified?

- Lines 475-482: Methods for DNA extraction and variant calling are not described. Link <http://www.agrf.org.au/services/genotyping> does not work.

- Lines 486-490: How many biological replicates were used? How were they formed? For instance, did the biological replicates correspond to flowers from 3 different inflorescences from the same plant? If "n = 3" corresponds to 3 technical replicates, gene expression analysis should be repeated using at least 3 biological replicates per flower sex type.

- Lines 500-507: Were the methods for cloning VviPLATZ1 for 03C003V0060 (M/f) the same as for 04C023V0006 (H/H) and 04C023V0003 (f/f)? If so, how the M and f alleles of VviPLATZ1 were distinguished?

Minor reviews

- Line 19: SEX-DETERMINING REGION

- Line 91: candidate sex-determining genes

- Line 119: "a role in pollen development"

- Line 128: "reduced expression of this gene correlates with the formation of female flowers"

- Lines 246-248: "Of the seven genes located in the female flower identity region, transcripts for VviPLATZ1 appeared to correlate with the formation of female flowers". Transcript abundance (or gene expression) of VviPLATZ1 appeared to correlate with reflex stamens.

- Line 273: female-specific

- Line 289: Tissue-specific

- Lines 322-323: "At this time point, VviPLATZ1 transcripts are significantly higher in 04C023V0006 flowers compared to 04C023V0003 flowers". Indicating the flower sex type of the individuals used in the study at each occurrence would help the reader. In addition, VviPLATZ1 transcripts are

significantly more abundant (or gene expression/transcript abundance of VviPLATZ1 is significantly higher) in 04C023V0006 hermaphroditic flowers compared to 04C023V0003 female flowers.

- Lines 490-491: "Total RNA was extracted from 100 mg of 03C003V0060 (M/f) at eight and six WPA, as well as leaves, as described above." 100 mg of 03C003V0060 (M/f) flowers?

Reviewer #2:

None

Reviewer #3:

Remarks to the Author:

The changes made to the manuscript "A major determinant of grapevine flower sex identity is controlled by the VviPLATZ1 transcription factor", including the title greatly improves the manuscript quality and clarity. In the revision phase, the authors answered all the reviewers questions and made the necessary changes to the manuscript.

Therefore, we are very pleased with the responses, the changes and the results presented in the manuscript. Moreover, we would like to congratulate the authors on a very interesting research paper.

This article was reviewed by Margarida Rocheta and João Lucas Coito from Instituto Superior de Agronomia, Lisbon, Portugal

The manuscript was reviewed by Margarida Rocheta and Lucas Coito

REVIEWER COMMENTS

Reviewer #1 (Remarks to the Author):

Main comments

As we mentioned in our past review, the results presented in the manuscript show elegantly that VviPLATZ1 plays a role in stamen architecture. However, the results do not show that VviPLATZ1 is involved in sex determination in grapes. Although reflex stamen is a morphologic trait that is commonly used to evaluate grape flower sex phenotype, female flowers with erect filaments were described in previous studies (Stout, 1921; Cunha et al., 2020; IPGRI, UPOV, OIV, 1997), as well as in this manuscript (“1-5% of the stamens initiated in f/f were stunted and grew to an average length of 1.2”), indicating that reflex stamens are not responsible for male sterility in Vitis female flowers. Indeed, self-pollination could still occur through wind or pollinators if the pollen grains were able to germinate. Male fertility could be tested using an in vitro pollen germination assay or by bagging inflorescences before blooming. As the effect of the loss of VviPLATZ1 function on male fertility was not evaluated in this study, the role of VviPLATZ1 in sex determination cannot be stated. Accordingly, we ask again the authors to revise the claim that VviPLATZ1 is a sex-determining gene.

References:

Stout (1921) Stout AB. Types of flowers and intersexes in grapes with reference to fruit development. New York Agricultural Experiment Station Technical Bulletin. 1921;82:1–16.

Cunha, J., Ibáñez, J., Teixeira-Santos, M., Brazão, J., Feveiro, P., Martínez-Zapater, J. M., & Eiras-Dias, J. E. (2020). Genetic Relationships Among Portuguese Cultivated and Wild *Vitis vinifera* L. Germplasm. *Frontiers in plant science*, 11, 127. <https://doi.org/10.3389/fpls.2020.00127>

IPGRI, UPOV, OIV. 1997. Descriptors for Grapevine (*Vitis* spp.). International Union for the Protection of New Varieties of Plants, Geneva, Switzerland/Office International de la Vigne et du Vin, Paris, France/International Plant Genetic Resources Institute, Rome, Italy. https://www.ecpgr.cgiar.org/fileadmin/bioersity/More_pubs/393_Descriptors_for_grapevine_Vitis_spp..pdf

We agree with the reviewer #1 that we have not shown a role for VviPLATZ1 in pollen infertility for sex determination. However, we have shown that loss of VviPLATZ1 is a major factor that determine female flower sex identity, as the reflex stamen phenotype is a key morphological trait associated with female flower formation. We have read through the manuscript and made additional changes to prevent any confusions about the loss of VviPLATZ1 in female flower formation.

Line 19: Changed “new candidate sex determinant” to “new candidate flower sex identity factor”

Line 130-131: Changed “new candidate molecular sex determinant” to “new candidate flower sex identity factor”

Lines 335: Changed “Based on the molecular and gene expression studies described above, we propose that *VviPLATZ1* is a new genetic determinant whose reduced function is a major factor of female flower sex identity” to “Based on the molecular and gene expression studies described above, we propose that reduced *VviPLATZ1* function is a major factor that determines female flower sex identity”.

Further, in light of this comment from reviewer #1, we have highlighted the importance of our work by discussing the fact that dioecious mating systems evolved to promote outcrossing, which not only is achieved by the failure to develop fertile pollen and ovaries, but also through the arrested or altered development of male and female reproductive organs. Please refer to the following lines: (A) 125-126

and (B) 390-396. The formation of heterostyly flowers is a good example of importance of the spatial separation of anthers of pistils for cross-pollination. As reviewer #1 points out above, self-pollination can still occur via wind and pollinators. The dependence of a pollination system on wind and pollinators would also result in cross-pollination. Therefore, the spatial separation of anthers and stigmas via reflex stamen development is likely to increase the outcrossing potential of the grapevine dioecious mating system.

Major reviews

Title

Title is confusing and misleading. It implies that *VviPLATZ1* controls a major determinant of grapevine flower sex determination. Authors use the term “determinant” for both genetic/molecular determinant and morphological determinant (line 123 for instance) in the manuscript, which is confusing. We suggest using this term for genetic determinant only. Also, the work doesn't implicate *VviPLATZ1* in sex determination, but in stamen morphology and the title should reflect that.

We have changed the title to “***VviPLATZ1* is a major factor that controls female flower sex identity in grapevine**” to increase clarity for the role of this gene in regulating the formation of female flowers. We have removed “sex determinant” in the manuscript (see above).

Results

- Line 149-150: “Results showed that the 5’ boundary of the SDR that specifies female flower identity was located in TPP, as indicated by the *VvMT_54* SNP (Fig. 1b).” Figure 1 a & b depict the *f* haplotype of Cabernet Sauvignon, but SNP calling was performed using the PN40024 haploid genome as reference. As previously mentioned, PN40024 genome is not appropriate as reference for studying the SDR. The genome of PN40024 is not appropriate as reference for detecting SNPs associated with flower sex trait, because the sex-determining region on chr2 is a mixture of *H* and *f* haplotypes (see Zhou et al., 2019 Nature Plants).

Therefore, we ask the authors to use a *H* or *f* haplotype of the SDR region as reference for the SNP analysis. One option is to use Cabernet Sauvignon, which as described in Massonnet et al. (2020) is available on a permanent repository (<https://doi.org/10.5281/zenodo.3827985>). Another option is to use the female *syvestris* C1-2, whose reads, genome assembly, and gene annotation are available in ENA under the bioproject PRJEB37020” (Badouin et al., 2020).

We have modified the Results (lines 146-207) Methods to clearly state how we identified SNPs and performed the fine mapping, which aligns with the concern and suggestion by reviewer #1. (Lines 494-543)

Results (lines 146-207)

Genetic and molecular analysis of *VviPLATZ1*. As shown in Fig. 1a, association genetics indicates that the *FSDR* is comprised of 11 genes from *TPP* to *WRKY21*³¹. To further narrow down the number of genes that reside in the *FSDR*, 21 SNPs spanning the PN40024 *SDR* were identified (Supplemental Table 1). Three genetic mapping resources were genotyped with this *SDR* SNP set: (1) *Vitis vinifera* cultivars with known flower sex genotypes (*f/f*, *H/f* or *H/H*), (2) an F_1 00C001V0008 (*f/f*) x Ugni Blanc (*H/f*) mapping population³⁷ and (3) self-fertilized hermaphrodite microvine (*H/f*) progenies (S1 to S5). However, as the PN40024 *SDR* contains a mixture of *H*- and *f*-sequences, this reference genome was not suitable for fine genetic mapping of female flower sex traits, as gene content between the *H*- and *f*-haplotypes differs in the *SDR*^{26,30}. Genomic studies indicate that gene content and order in the *SDR* of *f*-haplotypes is conserved^{27,28,31}. Therefore, we posited that the previously well characterized Cabernet Sauvignon *f*-haplotype sequence²⁸ was a suitable reference for fine genetic mapping of the

FSDR by identifying the likely relative positions of the genic SNP markers developed in our study (Supplementary Table 1). By using this genetic mapping strategy, the genotype information derived from the *Vitis vinifera* cultivars indicated that the 5' boundary of the *FSDR* was located in *TPP*, at the *VvMT_54* SNP (Fig. 1b; Supplementary Table 2). Genotype data from the F₁ 00C001V0008 (*f/f*) x Ugni Blanc (*H/f*) individuals further supported the location of the 5' *FSDR* boundary in *TPP* (Supplementary Table 2). A single recombination event at *VvMT72* in the S5-microvine, 06C008V0018, defined the 3' *FSDR* boundary located in *FLAVIN-CONTAINING MONOOXYGENASE 3 (FMO3)*; Fig 1b; Supplementary Table 1, 2). Taken together, fine genetic mapping demonstrated that the *FSDR* contained seven genes: *VviINP1*, *Exostosin*, 3-ketoacyl-acyl carrier protein synthase III (*KASIII*), *VviPLATZ1*, *FMO1*, *FMO4* and *FMO2* (Fig. 1b).

Methods (Lines 494-543)

DNA marker development and fine genetic mapping of the *FSDR*. Sequences for the SSR markers, UDVO27, VVIB23, VMC3B10 and VMC6F1, flanking the previously identified *SDR*^{21-23,59}, were used to identify the genomic region of this locus in the 8X PN40024⁶⁰ and Pinot Noir⁶¹ genomes. The genomic sequences of annotated genes regularly spaced between UDVO27 and VMC6F1 in PN40024 and Pinot Noir were scanned for candidate SNPs using the basic local alignment search tool (<https://blast.ncbi.nlm.nih.gov>). Thirty-three candidate SNPs were identified in exons and/or introns of selected genes in the region from VIT_202s0025g04920 to VIT_202s0154g00230 (Supplementary Table 1). The Agena Bioscience MassARRAY platform (Agena Bioscience, San Diego, CA, USA) was used to validate the 33 SNPs by genotyping *Vitis vinifera* cultivars (*n*=33) and self-pollinated microvines (*n*=37) outlined in Supplementary Table 2. In addition, the F₁ 00C001V0008 x Ugni Blanc mapping population (*n*=101) was also genotyped with this *SDR* SNP set. The MassARRAY genotyping was performed at the Australian Genome Research Facility (AGRF; <https://www.agrf.org.au/genotyping>). In this procedure, DNA was isolated from immature leaves using the NucleoSpin® 96 Plant II DNA extraction kit according to the manufacturer's instructions (<https://www.mn-net.com/nucleospin-96-plant-ii-96-well-kit-for-dna-from-plants-740663.4>). PCR was used to amplify the 33 SNP-targeted regions in each of the DNA samples. After dephosphorylating excess nucleotides in the PCR reactions, a single base extension reaction was performed using an extension primer for each of the 33 SNPs. The extension reaction utilized a single mass-modified dideoxynucleotide complementary to each of the SNP alleles. Each SNP allele was identified by the distinct mass of the extension primer using MALDI-TOF mass spectroscopy analysis⁶². Allele calling was performed using the Agena Bioscience MassARRAY software, version 4.1.0.83. After MassARRAY genotyping, 21 SNPs were validated for fine genetic mapping of the *FSDR* (Supplementary Table 1). The PCR primers used to amplify the 21 SNP targeted regions, as well as the corresponding extension primers are shown in Supplementary Table 3.

To perform the fine genetic mapping, the positions of the 21 validated SNPs were identified in the *SDR* sequences of the Cabernet Sauvignon *H*- and *f*-haplotypes. This was achieved by aligning a 50 bp PN40024 genomic sequence containing each SNP to the Cabernet Sauvignon *H*- and *f*-haplotypes using the basic local alignment search tool (Supplementary Table 1). Sequence and gene organization of the Cabernet Sauvignon *f*-haplotype together with the genotype information shown in Supplementary Table 2 was used to delineate the boundaries of the *FSDR*.

- Lines 258-261: It would be pertinent to present the results of *VviPLATZ1* expression in flowers from the male individual 03C003V0060 in Figure 1c because *VviPLATZ1* seems to be lower expressed in the male flowers compared to the hermaphrodite ones (Supplementary Fig. 1). If so, it should be discussed.

The focus of this paper is on identifying and functionally validating the VviPLATZ1 as a female flower identity factor. Differences in the expression levels between hermaphrodite and male flowers provide no information on the loss of VviPLATZ1 function in reflex stamen development. Further, the hermaphrodite and male plants used in this expression analysis are derived from different genetic backgrounds, which likely possess different genetic modifiers that influence the expression of this gene. Therefore, we feel that it is appropriate to display the male VviPLATZ1 expression data in the supplementary data.

- Line 262: “female flower identity region of the SDR”. Floral organ-identity genes are the genes specifying floral organ identity. Therefore, “female flower identity region” should be named the female-associated region.

It is true that floral organ identity genes specify floral organ identity in a whorl specific manner during flower development. But it is unclear as to why this relates to changing the name “female flower identity region” to “female associated region”. To maintain consistency for the reader and the current naming of the sex locus as SDR, we changed the “female flower identity region” to Female-SDR (FSDR) and the male flower identity region to Male-SDR (MSDR) in the manuscript.

Methods

A. Lines 469-471: “Alignment of the 8X PN40014 and Pinot Noir SDR-DNA sequences identified 21 single nucleotide polymorphisms (SNPs) spanning this locus”. First, PN40024 genome should not be used for studying sex determination. Second, how was the alignment performed and how were SNPs identified?

B. Lines 471-473: “These SNPs were validated by comparing H- and f-haplotype sequences in Cabernet Sauvignon (Supplementary Table 2) previously described²⁸” In Supplementary table 2: “1 From <http://www.grapegenomics.com>. PN40024_Chr02_V2.1. 1 From <http://www.grapegenomics.com>. f-haplotype of Cabernet Sauvignon.” The number “1” indicates for both PN40024 and Cabernet Sauvignon genomes. How were the H and f haplotype compared? How were SNPs identified?

C. Lines 475-482: Methods for DNA extraction and variant calling are not described. Link <http://www.agrf.org.au/services/genotyping> does not work.

We have modified the methods section to address reviewers #1 concerns from above: A-C (Lines 494-543). First, we have clarified how we identified in the SNPs. Second, we addressed how we used the SDR sequences from the Cabernet Sauvignon H- and f-haplotypes. Thirdly, we clarified the written the procedure utilized by AGRF for the MassARRAY genotyping.

DNA marker development and fine genetic mapping of the FSDR. Sequences for the SSR markers, UDV027, VVIB23, VMC3B10 and VMC6F1, flanking the previously identified *SDR*^{21-23,59}, were used to identify the genomic region of this locus in the 8X PN40024⁶⁰ and Pinot Noir⁶¹ genomes. The genomic sequences of annotated genes regularly spaced between UDV027 and VMC6F1 in PN40024 and Pinot Noir were scanned for candidate SNPs using the basic local alignment search tool (<https://blast.ncbi.nlm.nih.gov>). Thirty-three candidate SNPs were identified in exons and/or introns of selected genes in the region from VIT_202s0025g04920 to VIT_202s0154g00230 (Supplementary Table 1). The Agena Bioscience MassARRAY platform (Agena Bioscience, San Diego, CA, USA) was used to validate the 33 SNPs by genotyping *Vitis vinifera* cultivars ($n=33$) and self-pollinated microvines ($n=37$) outlined in Supplementary Table 2. In addition, the F_1 00C001V0008 x Ugni Blanc mapping population ($n=101$) was also genotyped with this SDR SNP set. The MassARRAY genotyping was performed at the Australian Genome Research Facility (AGRF; <https://www.agrf.org.au/genotyping>).

In this procedure, DNA was isolated from immature leaves using the NucleoSpin® 96 Plant II DNA extraction kit according to the manufacturer's instructions (<https://www.mn-net.com/nucleospin-96-plant-ii-96-well-kit-for-dna-from-plants-740663.4>). PCR was used to amplify the 33 SNP-targeted regions in each of the DNA samples. After dephosphorylating excess nucleotides in the PCR reactions, a single base extension reaction was performed using an extension primer for each of the 33 SNPs. The extension reaction utilized a single mass-modified dideoxynucleotide complementary to each of the SNP alleles. Each SNP allele was identified by the distinct mass of the extension primer using MALDI-TOF mass spectroscopy analysis⁶². Allele calling was performed using the Agena Bioscience MassARRAY software, version 4.1.0.83. After MassARRAY genotyping, 21 SNPs were validated for fine genetic mapping of the *FSDR* (Supplementary Table 1). The PCR primers used to amplify the 21 SNP targeted regions, as well as the corresponding extension primers are shown in Supplementary Table 3.

To perform the fine genetic mapping, the positions of the 21 validated SNPs were identified in the *SDR* sequences of the Cabernet Sauvignon *H*- and *f*-haplotypes. This was achieved by aligning a 50 bp PN40024 genomic sequence containing each SNP to the Cabernet Sauvignon *H*- and *f*-haplotypes using the basic local alignment search tool (Supplementary Table 1). Sequence and gene organization of the Cabernet Sauvignon *f*-haplotype together with the genotype information shown in Supplementary Table 2 was used to delineate the boundaries of the *FSDR*.

- Lines 486-490: How many biological replicates were used? How were they formed? For instance, did the biological replicates correspond to flowers from 3 different inflorescences from the same plant? If "n = 3" corresponds to 3 technical replicates, gene expression analysis should be repeated using at least 3 biological replicates per flower sex type.

We have clarified the number of biological replicates used in our study (lines 545-557)

Expression and cloning of *VviPLATZ1*. Total RNA was extracted from 100 mg of flowers derived from at least three inflorescences from a single 04C023V0006 (*H/H*), 04C023V0003 (*f/f*) and 03C003V0060 (*M/f*) plant at eight, six and four WPA (Supplementary Fig. 8)³⁸, using the Spectrum Plant Total RNA Kit (Sigma-Aldrich Pty. Ltd., Sydney, NSW, Australia). After RNA extraction, samples were purified using the RNA Clean & Concentrator™ (Zymo Research, Irvine, CA, USA). Total RNA was also extracted and purified from 100 mg of leaf tissue derived from 4 immature leaves, approximately 25 cm² in size, from 04C023V0006, 04C023V0003 and 03C003V0060. Three biological replicates were used for analyzing *VviPLATZ1* transcript abundance in developing flowers at eight, six and four WPA, as well as leaves, for the 04C023V0006 and 04C023V0003. For 03C003V0060, only two biological replicates were used for analyzing *VviPLATZ1* expression in developing flowers at eight, six and four WPA, as well as leaves (Note: each biological replicate was derived from a single plant). We also specify the number of biological replicates used in figure legends.

- Lines 500-507: Were the methods for cloning *VviPLATZ1* for 03C003V0060 (*M/f*) the same as for 04C023V0006 (*H/H*) and 04C023V0003 (*f/f*)? If so, how the *M* and *f* alleles of *VviPLATZ1* were distinguished?

The methods for cloning *VviPLATZ1* from 03C003V0060 (*M/f*) has been added and was the same for the 04C023V0006 (*H/H*) and 04C023V0003 (*f/f*). (Lines 565-568).

“To clone *VviPLATZ1*, the B26 primer (GACTCGAGTCGACATCGATTTTTTTTTTTTTTTTTT) was used to prime cDNA synthesis in the 04C023V0006 (*H/H*), 04C023V0003 (*f/f*) and 03C003V0060 (*M/f*) cDNA samples derived from flowers at six WPA.”

We sequenced five *VviPLATZ1* clones from 03V003V0060 (*M/f*). Four out of five *VviPLATZ1* clones were derived from the male allele.

Minor reviews

- **Line 19:** SEX-DETERMINING REGION
-Changed to “SEX-DETERMINING REGION”
- **Line 94:** candidate sex-determining genes
-Changed to “candidate sex-determining genes”
- **Line 121:** “a role in pollen development”
-Changed to “play a role in pollen development”
- **Line 131-132:** “reduced expression of this gene correlates with the formation of female flowers”
-Changed to “reduced expression of this gene correlates with the formation of female flowers”
- **Lines 211:** “Of the seven genes located in the female flower identity region, transcripts for *VviPLATZ1* appeared to correlate with the formation of female flowers”. Transcript abundance (or gene expression) of *VviPLATZ1* appeared to correlate with reflex stamens.
-Changed to “transcript abundance for *VviPLATZ1* appeared to correlate with reflex stamens.”
- **Line 236:** female-specific
-Changed to “female-specific”.
- **Line 322:** Tissue-specific
-Changed to “**Tissues-specific**”.
- **Lines 325-327:** “At this time point, *VviPLATZ1* transcripts are significantly higher in 04C023V0006 flowers compared to 04C023V0003 flowers”. Indicating the flower sex type of the individuals used in the study at each occurrence would help the reader. In addition, *VviPLATZ1* transcripts are significantly more abundant (or gene expression/transcript abundance of *VviPLATZ1* is significantly higher) in 04C023V0006 hermaphroditic flowers compared to 04C023V0003 female flowers.

-Changed to “At this time point, transcript abundance of *VviPLATZ1* were significantly higher in 04C023V0006 hermaphrodite flowers compared to 04C023V0003 female flowers (Fig. 1c)”.
- **Lines 545-552:** “Total RNA was extracted from 100 mg of 03C003V0060 (*M/f*) at eight and six WPA, as well as leaves, as described above.” 100 mg of 03C003V0060 (*M/f*) flowers?

Expression and cloning of *VviPLATZ1*. Total RNA was extracted from 100 mg of flowers derived from at least three inflorescences from a single 04C023V0006 (*H/H*), 04C023V0003 (*f/f*) and 03C003V0060 (*M/f*) plant at eight, six and four WPA (Supplementary Fig. 8)³⁸, using the Spectrum Plant Total RNA Kit (Sigma-Aldrich Pty. Ltd., Sydney, NSW, Australia). After RNA extraction, samples were purified using the RNA Clean & Concentrator™ (Zymo Research, Irvine, CA, USA). Total RNA was also extracted and purified from 100 mg of leaf tissue derived from four immature leaves, approximately 25 cm² in size, from 04C023V0006, 04C023V0003 and 03C003V0060.

[Editor: Reviewer #2 states in Remark to Editor section that (s)he is satisfied with the revision.]

Reviewer #3 (Remarks to the Author):

The changes made to the manuscript “A major determinant of grapevine flower sex identity is controlled by the VviPLATZ1 transcription factor”, including the title greatly improves the manuscript quality and clarity. In the revision phase, the authors answered all the reviewers questions and made the necessary changes to the manuscript.

Therefore, we are very pleased with the responses, the changes and the results presented in the manuscript. Moreover, we would like to congratulate the authors on a very interesting research paper.

This article was reviewed by Margarida Rocheta and João Lucas Coito from Instituto Superior de Agronomia, Lisbon, Portugal

The manuscript was reviewed by Margarida Rocheta and Lucas Coito

Reviewers' Comments:

Reviewer #1:

Remarks to the Author:

This new version of the manuscript addresses most of our concerns. However, the abstract still focuses on sex determination instead of flower architecture. The abstract should be modified to focus on the role of VviPLATZ1 in stamen erectness in grapes.

Line 13 and line 17: "sex determinant"

Line 22-23: "Phenotype analysis showed that individual lines homozygous for the gene-edited alleles gave rise to female flowers". We suggest: "Phenotype analysis showed that individual lines homozygous for the gene-edited alleles gave rise to flowers with reflex stamens, a phenotype typically observed in female flowers".

NCOMMS-21-09506

Response to Reviewer #1

Reviewer #1 (Remarks to the Author):

This new version of the manuscript addresses most of our concerns. However, the abstract still focuses on sex determination instead of flower architecture. The abstract should be modified to focus on the role of VviPLATZ1 in stamen erectness in grapes.

Response: The title has been changed to **“VviPLATZ1 is a major factor that controls female flower morphology determination in grapevine”**. Throughout the manuscript, including the abstract, we refer to VviPLATZ as a factor that regulates female flower formation and reflex stamen development.

Line 13 and line 17: “sex determinant”

We have not made the change to line 13 as there are more than one sex determinants in dioecious plants. Sex determinant has been removed from line 17.

Line 22-23: “Phenotype analysis showed that individual lines homozygous for the gene-edited alleles gave rise to female flowers”. We suggest: “Phenotype analysis showed that individual lines homozygous for the gene-edited alleles gave rise to flowers with reflex stamens, a phenotype typically observed in female flowers”.

We have made the following change to lines 23-24 as requested by reviewer #1, but in the present tense: “Phenotype analysis shows that individual lines homozygous for the gene-edited alleles produce flowers with reflex stamens.”.